# Dis-inhibitory neuronal circuits can control the sign of synaptic plasticity

**Julian Rossbroich**[1,2]  **Friedemann Zenke**[1,2]

`{firstname.lastname}@fmi.ch`
[1] Friedrich Miescher Institute for Biomedical Research, Basel, Switzerland
[2] Faculty of Science, University of Basel, Basel, Switzerland

## Abstract

How neuronal circuits achieve credit assignment remains a central unsolved question in systems neuroscience. Various studies have suggested plausible solutions for back-propagating error signals through multi-layer networks. These purely functionally motivated models assume distinct neuronal compartments to represent local error signals that determine the sign of synaptic plasticity. However, this explicit error modulation is inconsistent with phenomenological plasticity models in which the sign depends primarily on postsynaptic activity. Here we show how a plausible microcircuit model and Hebbian learning rule derived within an adaptive control theory framework can resolve this discrepancy. Assuming errors are encoded in top-down dis-inhibitory synaptic afferents, we show that error-modulated learning emerges naturally at the circuit level when recurrent inhibition explicitly influences Hebbian plasticity. The same learning rule accounts for experimentally observed plasticity in the absence of inhibition and performs comparably to back-propagation of error (BP) on several non-linearly separable benchmarks. Our findings bridge the gap between functional and experimentally observed plasticity rules and make concrete predictions on inhibitory modulation of excitatory plasticity.

## 1 Introduction

How do neurons far away from the sensory periphery and motor output system update their connections to contribute to network computation meaningfully? This question, formally known as the "credit assignment problem," is one of the outstanding questions in systems neuroscience. Classic learning theories assume synaptic plasticity in the brain is mainly Hebbian, i.e., changes in a synapse's efficacy depend merely on the pre- and postsynaptic activity of the neurons it connects [1]. A plethora of experiments in different brain areas support the notion of Hebbian plasticity [2, 3] and diverse phenomenological plasticity models quantitatively capture observed synaptic plasticity dynamics [4–8]. Theories of reinforcement learning suggested further extensions of Hebbian learning to three-factor rules that account for reward modulation, whereby a global modulatory factor, e.g., dopamine, explicitly influences the sign of plasticity [9, 10]. Again, there is ample experimental evidence for the role of neuromodulators in gating plasticity in various brain areas [11, 12]. Yet, recent work argues that global neuromodulation is insufficient to learn complex function mappings in large networks [13, 14] and that more fine-grained control over the sign of plasticity is required, as in the BP algorithm used in deep learning [15–18]. This realization motivated several modeling studies on how biological networks could approximate BP [19–25]. One central assumption in virtually all of these models is that the sign of plasticity is explicitly modulated by a neuron-specific local error signal akin to the gradient-based update used in BP [26]. However, learning rules with explicit error

37th Conference on Neural Information Processing Systems (NeurIPS 2023).

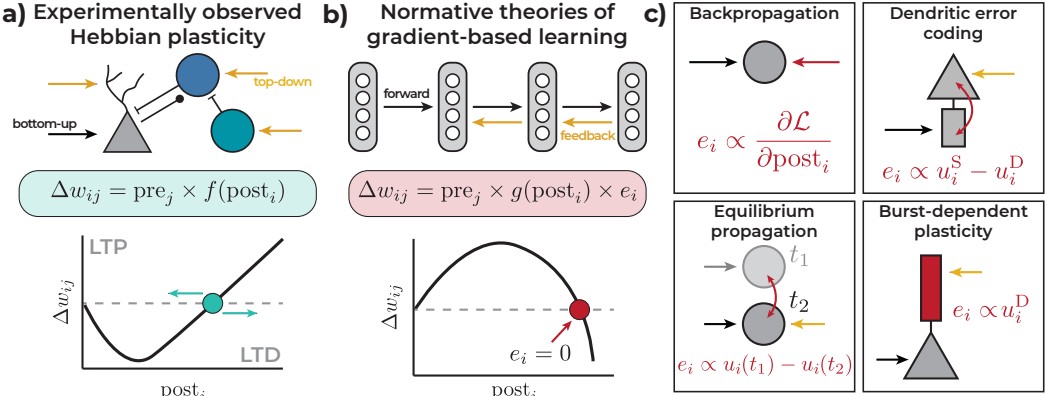

Figure 1: Explicit error modulation of the sign of plasticity is inconsistent with phenomenological plasticity models. **(a)** In neuronal circuits, top-down feedback connections target excitatory neurons, as well as inhibitory and dis-inhibitory circuits that have been implicated in gating of plasticity. In phenomenological plasticity models, the sign of plasticity is typically determined by postsynaptic quantities such as the membrane voltage [7], firing rate [6], or calcium concentration [8] without explicit error modulation. **(b)** In normative models, the sign of plasticity is typically subject to a hypothetical, explicit error modulation with little experimental evidence. Explicit error modulation makes specific predictions of the shape of the learning rule, (Supplementary Fig. S1; see Appendix A), at odds with experimentally observed plasticity (see Panel (a)). **(b)** Theories of bio-plausible gradient-based learning typically focus on approximating direct error-modulation as in BP, but differ in how errors are computed and relayed. Existing models suggest separate temporal phases, e.g. equilibrium propagation (EP) [29], putative compartments to represent error signals locally, e.g., dendritic error coding [22, 30], or burst-multiplexing [23, 31]. Still, there is little evidence for an explicit error signal that alters the sign of plasticity [26] and is consistent with phenomenological plasticity models.

modulation are inconsistent with phenomenological models of Hebbian plasticity (Fig. 1), raising the question of how theories of bio-plausible credit assignment tie into the phenomenology of Hebbian learning.

Here, we address this question using a normative control theory approach. We set out from a plausible dis-inhibitory circuit motif ubiquitously found in the brain [27, 28] and derive a Hebbian learning rule with an explicit inhibitory current dependence. We demonstrate that this learning rule accounts for key experimental observations *and* allows for control over the sign of plasticity through top-down synaptic input to specific interneurons. Our work suggests that error signals could naturally be encoded in top-down inputs to inhibitory neurons and bridges the gap between normative theories of gradient-based learning and phenomenological models of Hebbian plasticity. Our main contributions are:

- We extend Deep Feedback Control (DFC), a recent adaptive control theory framework for error-based learning [24], to a biologically plausible dis-inhibitory microcircuit motif.
- We show how a Hebbian plasticity rule with an explicit inhibition dependence can naturally decode credit signals from this microcircuit.
- We demonstrate that this rule enables error-based learning in hierarchical networks, naturally stabilizes runaway Hebbian plasticity, and resembles phenomenological plasticity rules under simulated experimental conditions.
- Finally, we demonstrate that our learning rule performs comparable to BP in deep neural networks trained on computer vision benchmarks.

## 2 Background and previous work

In this article we strive to reconcile phenomenological plasticity models constrained by experiments and models of biologically plausible credit assignment based on normative theories of gradient-based learning. In the following we review essential literature of both approaches.

## 2.1 Phenomenological synaptic plasticity models

There is widespread experimental support for the notion of Hebbian synaptic plasticity in the brain, captured in the form of classical long-term potentiation (LTP) and long-term depression (LTD) [2] or spike-timing-dependent plasticity (STDP) [3]. An extensive mathematical model catalog captures the phenomenology of these findings [4–8]. Common to most of the above models is that postsynaptic quantities define a plasticity threshold which separates LTD from LTP induction, which effectively determines the sign of the synaptic weight change [32–34] (Fig. 1a). A plethora of phenomenological plasticity models exist that capture such dependence on firing rate [6], voltage [7], and postsynaptic calcium concentration [5, 8]. For isolated neurons, classic work has demonstrated that Hebbian plasticity can extract principal components from structured data [35] or capture receptive field formation observed experimentally [36]. However, these models do not extend to deep hierarchical networks, nor can they account for the modulation of plasticity through local error signals required for solving the credit assignment problem. Thus, the mechanisms by which synaptic plasticity observed under experimental conditions could give rise to coordinated learning at the circuit- and network level remain elusive.

## 2.2 Models of biologically plausible credit assignment

The above models are contrasted by normative rules derived from gradient-based learning principles, which often aim at approximating BP [15] (Fig. 1b-c). A crucial aspect of BP is the separation of forward- and backward signaling, which algorithmically separates credit signaling from neuronal activity [37]. This separation presents a significant challenge for biologically plausible implementations, as it presumes a distinct separation through either learning phases or separate pathways.

Equilibrium propagation (EP) offers one possible, if only partial, solution to this dilemma. EP posits that local errors are derived as variations in neuronal activity at a network equilibrium state [29]. Yet, classic EP still requires separate processing phases for each input, inconsistent with neurobiology. However, recent work suggests possible ways of alleviating the requirement for distinct phases through neural oscillations [38].

An alternative to separate phases for forward and backward passes is to separate them spatially. Predictive coding models [19, 25, 39] exemplify this idea, whereby errors are computed in dedicated neuron-specific error units by comparing each neuron's activity to a top-down prediction. Recent work has suggested that the electrotonically segregated apical dendrites of cortical pyramidal neurons [37, 40] could represent local errors in learning [21, 22].

However, to approximate BP, a common theme across these models is their dependence on explicit error-modulation of plasticity [30] (Fig. 1b-c). While error-modulated learning rules prove functionally useful and, in some cases, can match the performance of BP, experimental evidence for their existence is inconclusive. In particular, they fall short of capturing established properties of experimentally observed plasticity, such as a threshold between LTD and LTP that is governed by postsynaptic activity [32–34, 41].

Finally, Payeur et al. [23] proposed a unique multiplexing approach by encoding forward and feedback signals in the event and burst rate of output spike trains. Such burst-dependent plasticity rules capture essential facets of phenomenological plasticity [31] (Supplementary Fig. S1; see Appendix A). However, the model primarily applies to cortical Layer 5 pyramidal cells with electrically segregated dendritic trees. In contrast, it may not work for layer 2/3 neurons or other brain areas without segregated dendrites.

Since most of the above models focused on approximating BP, they face another potential issue: they usually require weak feedback, which causes only a slight perturbation, or nudge, of the input-driven equilibrium. This requirement contrasts significantly with a wealth of experimental literature suggesting that feedback connections in the brain substantially influence neuronal activity [42–44]. Rather than striving to find biologically plausible separations between forward and backward signaling, recent modeling studies used an approach grounded in adaptive control theory [25, 45–48]. These models leverage strong feedback signals to steer neuronal activity to align with a given target output. While strong feedback aligns more closely with neurobiological observations, these models are also dependent on explicitly error-modulated learning rules, wherein the error is computed from either the difference between the controlled and uncontrolled states of neuronal activity similar to EP or the difference between activity in segregated neuronal compartments, as in dendritic error coding.

Still, it remains to be seen how such learning could be implemented at the circuit level with plasticity rules that capture experimental findings. In this article, we propose a putative circuit-level solution to this issue by mapping the notion of feedback control onto a known dis-inhibitory circuit motif and combining it with an inhibition-modulated Hebbian plasticity rule.

## 3 Model

To study whether biological microcircuits could naturally interpret dis-inhibitory feedback signals as local errors to control the sign of synaptic plasticity, we consider a continuous time dynamic multi-layer network comprised of excitatory and inhibitory neurons in each layer with dis-inhibitory feedback connections.

### 3.1 Neuronal dynamics

The membrane potential dynamics of the excitatory and inhibitory neurons in layer $i$ are described by the following ordinary differential equations (ODEs):

$$\tau_{\mathrm{E}}\frac{d}{dt}\mathbf{u}_i^{\mathrm{E}} = -\mathbf{u}_i^{\mathrm{E}}(t) + \mathbf{W}_i\mathbf{r}_{i-1}^{\mathrm{E}}(t) - \mathbf{r}_i^{\mathrm{I}}(t) \tag{1}$$

$$\tau_{\mathrm{I}}\frac{d}{dt}\mathbf{u}_i^{\mathrm{I}} = -\mathbf{u}_i^{\mathrm{I}}(t) + \mathbf{r}_i^{\mathrm{E}}(t) - \mathbf{Q}_i\mathbf{c}(t) \tag{2}$$

with $\mathbf{W}_i$ the afferent synaptic weights from the previous layer and $\mathbf{r}_i = \phi(\mathbf{u}_i)$ a smooth, monotonically increasing nonlinear activation function. Note that here we made the simplifying assumption that each excitatory neuron has an associated inhibitory neuron and that both are connected locally within a microcircuit. For all simulations, we use the soft rectifying nonlinearity $\phi(u) = \beta\log(1+\exp(u-\gamma))$, in which $\beta$ and $\gamma$ are parameters controlling the scale and shift of the activation function, respectively. Dis-inhibitory feedback is mediated through top-down control signals $\mathbf{c}(t)$ relayed to each layer $i$ through an associated feedback weight matrix $\mathbf{Q}_i$ (Fig. 2). The input layer $\mathbf{r}_0(t)$ is data-dependent and not influenced by top-down feedback. For a constant input current $\mathbf{r}_0$, the network dynamics settle to the equilibrium state

$$\overset{*}{\mathbf{u}}_i^{\mathrm{E}} = \mathbf{W}_i\overset{*}{\mathbf{r}}_{i-1}^{\mathrm{E}} - \overset{*}{\mathbf{r}}_i^{\mathrm{I}} \quad , \quad \overset{*}{\mathbf{u}}_i^{\mathrm{I}} = \overset{*}{\mathbf{r}}_i^{\mathrm{E}} - \mathbf{Q}_i\overset{*}{\mathbf{c}} \quad . \tag{3}$$

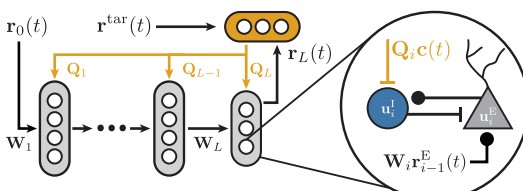

Figure 2: Illustration of a multi-layer network with dis-inhibitory control microcircuits (left). Each network unit consists of an excitatory and inhibitory neuron that are recurrently connected (right). The top-down control signal to each layer $\mathbf{Q}_i\mathbf{c}(t)$ is relayed through dis-inhibitory afferents (orange).

### 3.2 Feedback control

As in previous work on DFC [24, 47], our model uses feedback control to drive the output activity of the network towards the target activity $\mathbf{r}_L^{\mathrm{tar}}$ by minimizing the magnitude of the output error:

$$\mathbf{e}(t) = -\left.\frac{\partial\mathcal{L}(\mathbf{r}_L, \mathbf{r}_L^{\mathrm{tar}})}{\partial\mathbf{r}_L}\right|_{\mathbf{r}_L=\mathbf{r}_L(t)}^{T} \tag{4}$$

where $\mathcal{L}(\mathbf{r}_L, \mathbf{r}_L^{\mathrm{tar}})$ is a label-dependent supervised loss function defined on the network's output activity. For a simple mean squared error (MSE) loss, $\mathcal{L} = 1/n\sum_n 1/2\|\mathbf{r}_L^{\mathrm{tar}}(n) - \mathbf{r}_L(n)\|_2^2$, the error for each datapoint $n$ is directly related to the difference between the network's output and the target, specifically $\mathbf{e}(t) = \mathbf{r}_L^{\mathrm{tar}} - \mathbf{r}_L(t)$.

**Leaky proportional-integral controller.** In our model, we use a leaky proportional-integral controller to compute the feedback control signals $\mathbf{c}(t)$:

$$\mathbf{c}(t) = k_{\mathrm{p}}\mathbf{e}(t) + k_{\mathrm{i}}\mathbf{c}^{\mathrm{int}} \quad , \quad \tau_c\frac{d}{dt}\mathbf{c}^{\mathrm{int}} = \mathbf{e}(t) - \mathbf{c}^{\mathrm{int}}(t) \tag{5}$$

where $k_{\mathrm{p}}$ and $k_{\mathrm{i}}$ are the proportional and integral control constants, respectively.

**Dis-inhibitory feedback connectivity.** Next, we have to connect the controller to the controlled quantities. In the context of our model, we assume that control signals are relayed via inhibitory interneurons. We thus have to specify the feedback weights connecting the control signals to the local microcircuits defined in the previous section. While we make no claims about how suitable control feedback is generated in neurobiology, in this article we merely assume that suitable control signals exist and that they are mediated via inhibitory interneurons. To that end, the feedback weights $\mathbf{Q}_i$ need to be chosen such that the output loss $\mathcal{L}(\mathbf{r}_L, \mathbf{r}_L^{\text{tar}})$ is minimized at the controlled equilibrium state. To fulfill this requirement, the column space of the concatenated feedback weights of the network, $Q \triangleq \left[\mathbf{Q}_1^T, \ldots, \mathbf{Q}_L^T\right]^T$, must be equal to the row space of the network Jacobian $J$ at steady-state [24]. This Jacobian characterizes how infinitesimal perturbations of each controlled quantity, e.g., a neuronal activation, relates to changes in the network's output $\mathbf{r}_L$. In contrast to previous models relying on top-down control, our network consists of recurrently connected excitatory and inhibitory units whilst top-down input is targeting the inhibitory population exclusively. Since we want to model control through dis-inhibitory circuits, we assume that the Jacobian for the controller is defined with respect to the inhibitory membrane potential at each layer:

$$J \triangleq [\mathbf{J}_1, \ldots, \mathbf{J}_L] = \left[\frac{\partial \mathbf{r}_L}{\partial \mathbf{u}_1^{\text{I}}}, \ldots, \frac{\partial \mathbf{r}_L}{\partial \mathbf{u}_L^{\text{I}}}\right] \tag{6}$$

where we use $\frac{\partial \mathbf{r}_L}{\partial \mathbf{u}_i^{\text{I}}} = \nabla_{\mathbf{u}_i^{\text{I}}} \mathbf{r}_L$ to denote the Jacobian matrix of partial derivatives of the vector $\mathbf{r}_L$ with respect to the vector $\mathbf{u}_i^{\text{I}}$. It has been shown that a wide range of possible feedback weights $Q$ can match the row space of $J$ in the DFC framework. One simple way of ensuring this condition is met is to set the feedback weights at each layer proportional to the transposed Jacobian, i.e., $-\mathbf{Q}_i = \mathbf{J}_i^T$. However, the Jacobian depends on the input and the neuronal activity. It is thus changing over time until an equilibrium state is reached. To avoid changing feedback weights over time for a given input, we compute them based on the network Jacobian at the uncontrolled equilibrium state with $\mathbf{c} = 0$

$$-\mathbf{Q}_i = \tilde{\mathbf{J}}_i^T = \left(\frac{\partial \mathbf{r}_L}{\partial \mathbf{u}_i^{\text{I}}}\right)^T \bigg|_{\mathbf{u}_i^{\text{I}} = \tilde{\mathbf{u}}_i^{\text{I}}} \tag{7}$$

with $\tilde{\mathbf{u}}_i^{\text{I}}$ corresponding to the inhibitory membrane potentials in Layer $i$ at the uncontrolled equilibrium state.

**Learning as minimization of control in dis-inhibitory neuronal circuits.** Given suitable feedback weights and a strong influence on the network activity by the controller, neuronal activity can change considerably compared to the uncontrolled steady-state. Tracing the steps of [47], learning rules can be derived from a *minimization of control* objective

$$\mathcal{H} = \frac{1}{2} \sum_n \|Q\overset{*}{\mathbf{c}}(n)\|_2^2 \tag{8}$$

where $\overset{*}{\mathbf{c}}(n)$ is the steady-state feedback control signal for datapoint $n$. Formally, it can be shown that minimizing the above surrogate loss $\mathcal{H}$ also minimizes the output loss $\mathcal{L}$ (see Appendix B.1). We start with the following learning rule which minimizes $\mathcal{H}$:

$$\tau_w \frac{\mathrm{d}}{\mathrm{d}t}\mathbf{W}i = \Big[\underbrace{\big(\overset{*}{\mathbf{r}}_i^{\text{E}} - \overset{*}{\mathbf{u}}_i^{\text{I}}\big)}_{\text{error}} \odot \underbrace{\phi'\big(\overset{*}{\mathbf{u}}_i^{\text{E}}\big)}_{\text{postsynaptic}}\Big] \underbrace{\big(\overset{*}{\mathbf{r}}_{i-1}^{\text{E}}\big)^T}_{\text{presynaptic}} \tag{9}$$

where $\phi'(\mathbf{u})$ is the derivative of the activation function and $\odot$ denotes element-wise multiplication. The sign of the weight change is determined by the error projected onto each neuron by the feedback controller at the equilibrium, which following Eq. (3) is encoded as $\mathbf{Q}_i\overset{*}{\mathbf{c}} = \overset{*}{\mathbf{r}}_i^{\text{E}} - \overset{*}{\mathbf{u}}_i^{\text{I}}$. While Eq. (9) minimizes $\mathcal{H}$ when provided with sensible feedback signals, it does not constitute a local learning rule because it explicitly depends on the inhibitory membrane potentials $\overset{*}{\mathbf{u}}_i^{\text{I}}$, which excitatory neurons cannot access directly (cf. Fig. 1).

### 3.3 A Hebbian learning rule for error-modulated learning through dis-inhibitory control

We were wondering whether excitatory neurons could compute an effective local approximation of Eq. (9) by estimating the inhibitory membrane potentials from locally available quantities. To that

end, we first re-write Eq. (9) as

$$\tau_W \frac{\mathrm{d}}{\mathrm{d}t}\mathbf{W}_i = \left[\left(\overset{*}{\mathbf{r}}_i^{\mathrm{E}} - \phi^{-1}\left(\overset{*}{\mathbf{r}}_i^{\mathrm{I}}\right)\right) \odot \phi'\left(\overset{*}{\mathbf{u}}_i^{\mathrm{E}}\right)\right]\left(\overset{*}{\mathbf{r}}_{i-1}^{\mathrm{E}}\right)^T \quad, \tag{10}$$

where we substituted the inhibitory membrane potentials $\overset{*}{\mathbf{u}}_i^{\mathrm{I}}$ with the inverse activation function $\phi^{-1}\left(\overset{*}{\mathbf{r}}_i^{\mathrm{I}}\right)$. Mathematically, Eqs. (9) and (10) are equivalent, but conceptually, it re-frames the problem of non-locality since it would require an excitatory neuron to compute the inverse activation function from the recurrent inhibitory current, which is locally available. While it is hard to imagine how neurons would invert the activation function of other neurons exactly, we assume that they could conceivably compute a linear approximation, leading to a learning rule of the following general form:

$$\tau_W \frac{\mathrm{d}}{\mathrm{d}t}\mathbf{W}_i = \left[\left(\overset{*}{\mathbf{r}}_i^{\mathrm{E}} - \theta_i - \delta_i\overset{*}{\mathbf{r}}_i^{\mathrm{I}}\right) \odot \phi'\left(\overset{*}{\mathbf{u}}_i^{\mathrm{E}}\right)\right]\left(\overset{*}{\mathbf{r}}_{i-1}^{\mathrm{E}}\right)^T \quad. \tag{11}$$

Here the parameter $\theta_i$ takes the role of a postsynaptic plasticity threshold, common to many phenomenological plasticity models [6–8], while $\delta_i$ adds an inhibitory current dependence to this threshold. In practice, we obtain $\theta_i$ and $\delta_i$ through a first-order Taylor expansion of the inverse activation function around a given linearization point $\tilde{r}$ (see Appendix B). In the next sections, we will see that, depending on the inhibitory activation function and the linearization parameters, the model reconciles aspects of phenomenological plasticity with an effective error-modulation mechanism as demanded by normative theories of gradient-based learning.

## 4 Learning with dis-inhibitory control accounts for key plasticity experiments

Most experiments on synaptic plasticity are performed *in vitro* under highly controlled conditions, in which pairs of connected neurons are isolated. This enables researchers to investigate the plasticity at single synapses in the absence of interfering activity from the local microcircuit or long-range synaptic afferent connections. Such experimental conditions would likely interfere with any putative error-modulation of synaptic plasticity since long-range synaptic afferents are either severed during sample preparation or do not transmit plausible activity levels. To account for such possible experimental confounding factors and to compare experimentally observed Hebbian plasticity, we probe our error-modulated learning rule in three different settings: full microcircuit with "closed-loop feedback", intact "microcircuit without feedback", and isolated neurons resembling in-vivo experimental conditions with "direct control of inhibition" (Fig. 3). In each setting, we vary the excitatory input to the excitatory neuron and calculate the resulting weight change as a function of the postsynaptic firing rate using Eq. (11). For a qualitative comparison to previously suggested error-driven plasticity rules, we evaluated learning rules using dendritic error coding [22, 24, 30] or burst-dependent plasticity [23, 31] in comparable settings (Supplementary Fig. S1; see Appendix A).

**Dis-inhibition controls the sign of plasticity in the intact microcircuit.** We first considered a simplified version of the intact microcircuit with top-down dis-inhibitory feedback, for which our learning rule was derived. In this circuit, top-down control drives neuronal activity towards a target value $r^{\mathrm{tar}}$. We computed the weight updates dictated by our learning rule for two different targets as a function of the neuronal firing rate. In this setting, the learning rule exhibits two stable fixed points separated by an unstable one (Fig. 3a). Importantly, a stable fixed point exists at the the target firing rate, i.e. $r^{\mathrm{E}} = r^{\mathrm{tar}}$ (Fig. 3a). At this fixed point, the sign of plasticity is determined by the top-down controller through the error decoded by the learning rule (11). This behavior is in line with the behavior of learning rules with explicit error-modulation in this simplified setting (cf. Fig. 1b, Supplementary Fig. S1).

Moreover, in contrast to purely error-modulated plasticity rules, our rule exhibits an LTD region flanked by a stable fixed point in neuronal firing rates at zero and an unstable fixed point at intermediate firing rates. The existence of this LTD regime is a direct consequence of the local approximation used in its derivation (cf. Eq. (11)) and depends on the chosen parameters of the linearization (Supplementary Fig. S2; see Appendix B). For proper error-modulated learning in our framework, we have to ensure that each neuron's activity does not exclusively stay in this region over time and across different inputs. In neurobiology, this activity regime could, for instance, be attained through homeostatic plasticity [51]. Thus our learning rule exhibits error-modulated plasticity at the upper fixed point when embedded in an intact microcircuit with top-down feedback.

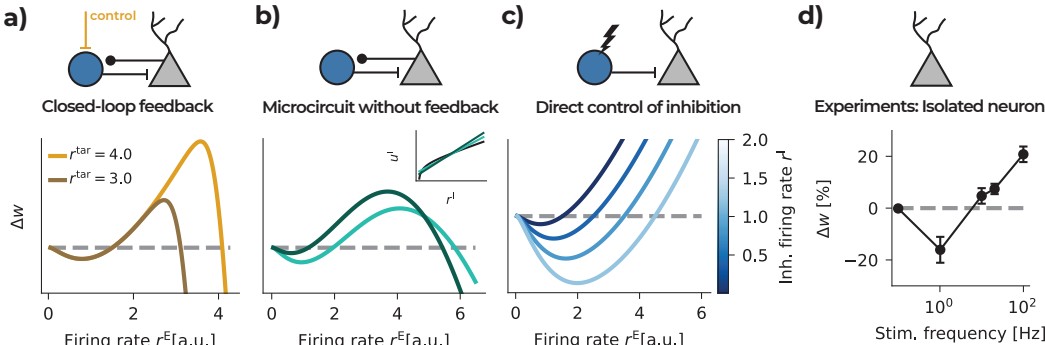

Figure 3: A Hebbian learning rule for error-modulated learning through dis-inhibitory control resembles plasticity observed in single-neuron electrophysiology experiments. **(a)** Weight change $\Delta W$ of a a single synapse as a function of postsynaptic firing rate $r_{\mathrm{E}}$. Different colored lines indicate two different postsynaptic firing rate targets $r^{\mathrm{tar}}$ indicated by colored arrows. The top-down feedback onto the interneuron is proportional to the error $r^{\mathrm{tar}} - r^{\mathrm{E}}$. In the intact microcircuit with closed-loop feedback, our learning rule naturally leads to error-modulated learning. **(b)** Same as (a), but with top-down connections ablated. The two shades of green represent two different linear approximations of the inverse inhibitory activation function (see inset; inverse activation function in black). The plasticity rule resembles a multi-stable Hebbian plasticity rule [49]. **(c)** Same as before, but for an isolated neuron without microcircuit. Different shades of blue correspond to different amounts of inhibitory current. Different levels of injected inhibitory current lead to differnt values for the plasticity threshold, but the stable fixed point disappears in the absence of recurrent inhibition. **(d)** Experimentally observed plasticity with activity-dependent LTD and LTP redrawn from [50]. The data qualitatively resembles our learning rule in the open-loop setting (cf. panel (c)).

**Plasticity in an isolated microcircuit is self-stabilizing.** To examine how our learning rule behaves in the absence of top-down control signals, we investigated plasticity dynamics in an isolated microcircuit without control feedback, while local circuit connectivity between excitatory and inhibitory neurons was left intact. In this setting, the inhibitory membrane potential does not encode an error signal that can be decoded by the learning rule. While explicitly error-modulated learning rules would not exhibit any synaptic weight change in this setting, our Hebbian learning rule predicts weight changes due to the imperfect approximation of the inverse function. Fig. 3b depicts the resulting plasticity dynamics for two different linear approximations. Embedded in a local microcircuit, the plasticity rule still exhibits both an LTD and LTP regime and a stable fixed point that depends on the learning rule parameters. Notably, the plasticity rule embedded in an isolated, but intact microcircuit is self-stabilizing through recurrent inhibition. Interestingly, the necessity of such a stable fixed point at higher activity levels for stable learning has been postulated previously in theoretical work [49, 52]. As before, the presence and location of the stable fixed point depend on the choice of plasticity parameters (Supplementary Fig. S2).

**Plasticity induction changes under direct control of inhibition.** Experiments on excitatory plasticity *in vitro* are commonly performed under conditions designed to minimize the interference of inhibitory activity, for example by applying GABA antagonists [53]. To study plasticity induction in our model under such simulated experimental conditions, we blocked recurrent connections from excitatory to inhibitory neurons, so that inhibitory activity is independent of excitatory activity. Additionally, we controlled inhibitory activity, as could be achieved, for instance, through current injection or optogenetic manipulations in experiments. In the absence of any inhibitory activity, i.e. $r^{\mathrm{I}} = 0$, our learning rule reduces to the form $\Delta w \propto r_{\mathrm{pre}} \left( r_{\mathrm{post}} - \theta \right) \phi' \left( r_{\mathrm{post}} \right)$ and loses its stable fixed point (Fig. 3c). Thus, in the absence of inhibition, the weight update prescribed by our plasticity model resembles Hebbian plasticity rules commonly observed under experimental conditions (Fig. 3d). However, our model predicts that direct control over inhibitory inputs to the excitatory neuron should shift the postsynaptic plasticity threshold to larger values.

In summary, dis-inhibitory control accounts for key plasticity experiments while also supporting error-driven learning in top-down controlled microcircuits. Additionally, its self-stabilizing capabilities provide a possible explanation as to why *in vitro* experiments have failed to uncover a

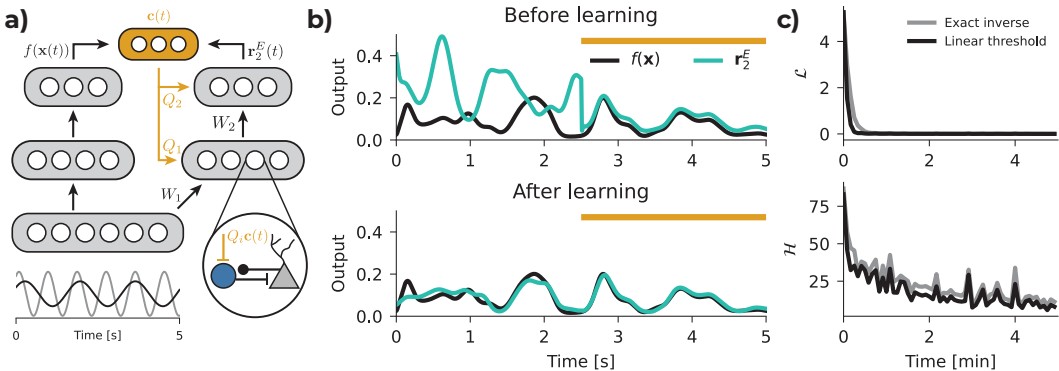

Figure 4: Online learning using dis-inhibitory control of Hebbian plasticity in a student-teacher task. **(a)** A teacher network (left) of size 30-20-2 implements a nonlinear mapping from an array of input sine waves $\mathbf{x}(t)$ (bottom) to an output target $f(\mathbf{x}(t))$. A student network (right) of the same size learns to approximate the teacher function by minimizing the control signal $\mathbf{Q}_i\mathbf{c}(t)$ of a dis-inhibitory feedback controller at each layer. **(b)** One example teacher output neuron (black) and the corresponding student neuron (teal) before and after learning. The yellow bar indicates when feedback control is active. **(c)** MSE loss $\mathcal{L}$ and least control loss $\mathcal{H}$ over time for student networks trained with the exact update rule derived in Eq. (10) and the linear threshold rule (Eq. (11)).

stabilizing mechanism for excitatory synaptic plasticity. Next, we test whether our learning rule allows hierarchical networks to solve nonlinear function approximation problems.

## 5 Dis-inhibitory control orchestrates learning in multi-layer networks

To explore our model's ability to train multi-layer networks, we first designed a simple continuous-time low-dimensional student-teacher learning task (Fig. 4a; see Appendix C for details). In this task, a randomly initialized teacher network with fixed parameters $f(\mathbf{x}(t))$ is given a set of 20 sine wave inputs with randomly chosen amplitudes, frequencies, and phases $\mathbf{x}(t)$. An architecturally identical student network but with different initial weights receives the same input $\mathbf{x}(t)$ and is tasked to reproduce the teacher's output, i.e. $r_L^{\mathrm{tar}}(t) = f(\mathbf{x}(t))$, evaluated through an MSE loss function. A dis-inhibitory feedback controller as described in the previous section is continually driving the student network's activity towards lower loss in real time. We trained the student network with either the exact non-linear inhibitory threshold rule Eq. (10) or the Hebbian learning rule with a linear inhibitory threshold Eq. (11). Neuronal and weight dynamics were simulated in continuous time using an explicit 5$^{\mathrm{th}}$ order Runge-Kutta method. To make sure that the feedback controller is aligned with the changing Jacobian during learning, the feedback weights were plastic and continuously evolving towards the average network Jacobian (see Appendix C).

Before learning, the student network did not follow the target closely in the open-loop setting, i.e., when the control signal was turned off. However, as soon as dis-inhibitory control was activated, the output activity closely followed the target (Fig. 4b). We then trained the student network for a total of 300 seconds of continuous sine wave inputs using either Eq. (10) or Eq. (11). After learning, the student network output closely followed the target in the open-loop setting, accompanied by a substantial reduction of the open-loop MSE loss ($c(t) = 0$) and the surrogate loss $\mathcal{H}$ over the course of training (Fig. 4c). Finally, we observed that the linear threshold learning rule resulted in comparable performance to the exact inverse learning rule. Thus, an inhibitory modulated Hebbian learning rule is capable of solving a nonlinear learning task when credit is relayed through local dis-inhibitory microcircuits acting as feedback controller.

### 5.1 Training multi-layer networks on classification tasks through dis-inhibitory control

Having confirmed that our learning rule is capable of error-driven learning through minimization of control on a simple continuous-time learning task, we wondered whether we could train a multi-layer perceptron (MLP) on standard image classification tasks. To that end, we implemented a MLP with excitatory-inhibitory microcircuit units. Networks were comprised of either one or three hidden layers

and used a parameterized soft rectifier activation function for all units. Numerical integration was performed using a fifth-order Runge-Kutta method with an adaptive step size to allow the network to reach an equilibrium state for each input (see Appendix C for details). Networks were trained either with the exact inverse rule in Eq. (10), or the linear threshold rule (Eq. (11)). To make training more robust and less dependent on initialization, the linear approximation of the inverse function was obtained through first-order Taylor expansion around a neuron-specific parameter $\tilde{\mathbf{r}}_i$ which tracked the average inhibitory firing rate across each batch (see Appendix C). With these settings we trained the networks on MNIST [54] and Fashion-MNIST [55]. For comparison, we also trained conventional MLPs with the same neuron numbers but strictly feed-forward connectivity using BP.

Table 1: Test accuracy in % for networks trained with BP or dis-inhibitory feedback control. Reported values are mean $\pm$stdev ($n = 10$). For validation accuracy see Appendix C.

| | | MNIST | | Fashion-MNIST | |
|---|---|---|---|---|---|
| Number of hidden layers | | 1 | 3 | 1 | 3 |
| Backprop | | $98.1 \pm 0.2$ | $98.3 \pm 0.1$ | $89.3 \pm 0.3$ | $89.4 \pm 0.2$ |
| Dis-inhibitory control | Exact inverse | $97.7 \pm 0.2$ | $98.0 \pm 0.2$ | $89.1 \pm 0.2$ | $89.1 \pm 0.2$ |
| | Linear threshold | $97.1 \pm 0.1$ | $96.5 \pm 0.1$ | $87.6 \pm 0.4$ | $86.8 \pm 0.2$ |
| | Exact inverse (Avg $\mathbf{J}$) | $97.8 \pm 0.1$ | $97.0 \pm 1.5$ | $88.7 \pm 1.8$ | $88.2 \pm 0.5$ |
| | Linear threshold (Avg $\mathbf{J}$) | $96.5 \pm 0.1$ | $96.0 \pm 0.3$ | $86.4 \pm 0.3$ | $84.6 \pm 0.3$ |

We observed that networks trained using dis-inhibitory control with the exact inverse learning rule performed almost on par with standard BP on both datasets (Table 1). Using the linear threshold rule led to a slight drop in accuracy in all cases. It is possible that this gap could be narrowed further by choosing different linearization parameters $\tilde{\mathbf{r}}_i$ since the performance of the exact inverse rule suggests that improving the approximation of the error translates to higher accuracy.

In above simulations, the Jacobian used for calculating the feedback signals is input-dependent and feedback weights are thus different for each stimulus. In neurobiology, feedback would presumably be relayed via synaptic connections that do not change this rapidly. To test whether successful learning is possible with more slowly varying feedback weights we repeated the above simulations with feedback weights that slowly tracked the average Jacobian (see Appendix C). This change did not compromise learning, although it resulted in a small but noticeable drop in accuracy compared to the ideal data-dependent Jacobian (Table 1). In summary, the combination of dis-inhibitory control with a Hebbian learning rule with an inhibition-dependent threshold allows training MLPs on vision datasets such as MNIST and Fashion-MNIST to accuracy values close to networks trained with BP.

## 6  Discussion

In this article, we introduced a Hebbian learning rule with an inhibition-dependent threshold, which, through dis-inhibitory microcircuit dynamics, allows top-down feedback to control the sign of plasticity. Notably, the learning rule captures essential aspects of classic phenomenological Hebbian plasticity models under simulated experimental conditions disrupting recurrent inhibition in the local microcircuit. In contrast to standard Hebbian plasticity models, our model is stable when recurrent inhibition is intact without requiring additional homeostatic or compensatory mechanisms. Finally, we show how dis-inhibitory control is sufficient to train MLPs on vision tasks with performance levels close to classical BP.

**Dis-inhibitory control reconciles error-based learning with classic Hebbian plasticity models.** The learning rule we put forward in this article has a postsynaptic plasticity threshold $\theta$ (cf. Eq. (11)), a neuron-specific parameter potentially subject to its own temporal dynamics. This threshold makes it reminiscent of the BCM rule [36]. Moreover, our model extends classic plasticity models by an explicit dependence on inhibition, which adds a dynamic, rapidly evolving component to the plasticity threshold. We derived this dual-threshold mechanism within a normative minimization-of-control objective. Its functional role is to decode a credit signal, relayed through dis-inhibitory afferents, from changes in the inhibitory current. Notably, the inhibitory threshold in our model confers an additional helpful property in that it induces a rapid compensatory plasticity mechanism that prevents

pathological runaway LTP, usually associated with Hebbian plasticity, through recurrent inhibition and in the absence of top-down control signals or any additional homeostatic mechanisms. Theoretical studies have argued that such a stabilizing mechanism for high firing rates should exist to counteract the runaway potentiation of Hebbian plasticity [49, 52].

**Experimentally testable predictions.**   Our model puts forth several testable predictions. First, we anticipate a direct modulation of plasticity induction through inhibitory currents in conventional excitatory long-term plasticity induction protocols. We predict that postsynaptic plasticity thresholds [32, 33] should be influenced by inhibitory current injection, even when the postsynaptic activity is kept constant in experiments. Most plasticity experiments proceed in the presence of GABA antagonists or sodium channel blockers, which may obscure the direct impact of inhibition on the plasticity threshold. Nevertheless, abundant experimental evidence supports the idea that inhibition influences classic induction protocols at excitatory synapses [56–59] and that GABAergic afferents can switch the sign of plasticity [60, 61]. Second, our model suggests that blocking dis-inhibitory circuits during an error-based learning paradigm should block or influence learning. Consistent with this hypothesis, dis-inhibitory microcircuits have been implicated with the gating of plasticity and behaviorally relevant learning in the Amygdala [62, 63], Hippocampus [64], and sensory cortices [65–69] (for a review, see [70]). Conversely, activating the same circuitry should affect or trigger learning during specific tasks.

**Limitations.**   Several limitations should be considered when interpreting this study's results. First, while our learning rule minimizes the loss, it does not necessarily follow the negative gradient. This difference could lead to sub-optimal learning dynamics and we will explore its impact in future work.

Moreover, our circuit model requires specific one-to-one connectivity between excitatory and inhibitory interneurons that is inconsistent with circuit motifs observed in the brain. In biological microcircuits, excitatory neurons usually exceed the number of inhibitory neurons [71], and inhibitory interneurons typically provide inputs to many local excitatory cells and vice versa. Nevertheless, it may be possible to consider our model's inhibitory threshold (cf. Eq. (11)) as a local inhibitory current derived from multiple presynaptic targets. In this scenario, the challenge for excitatory neurons is to estimate their respective contribution to the inhibitory current they receive. In future work, we will explore the possibility of learning an estimate of the expected inhibitory current by implementing inhibitory plasticity to achieve an excitatory-inhibitory balanced state [72] and its co-dependence with excitatory plasticity [73]. Inhibitory plasticity and its interactions with excitatory plasticity is supported by experiments [74–76] and has been the focus of recent computational models [77, 78]. However, neither of these studies explored the functional relevance of co-dependent plasticity for credit assignment and circuit-level learning.

Finally, the present model does not adhere to Dale's law in synaptic connections between hidden layers or the feedback pathway. Instead, we integrated the local population of dis-inhibitory interneurons into the dynamics of a top-down controller that modifies inhibitory activity bidirectionally. Biologically, this could be achieved through high baseline firing rates of dis-inhibitory interneurons or top-down feedback connections that target inhibitory interneurons directly [42]. In future work, we will explore detailed cortical architectures for dis-inhibitory feedback controllers that allow bi-directional modulation.

In summary, we made the first step to reconcile realistic circuit models and phenomenological plasticity rules with normative theories relying on error-modulated plasticity to solve the credit assignment problem. Specifically, we showed how top-down feedback signals targeting specific interneurons could efficiently modulate neuronal activity *and* plasticity. Our results highlight the potential of learning algorithms beyond BP, showcase their ability to incorporate diverse plasticity phenomena observed in neurobiology, and open the door for exciting future research.

## Acknowledgments

We thank all members of the Zenke Group for valuable comments and discussions. This project was supported by the Swiss National Science Foundation [grant number PCEFP3_202981] and the Novartis Research Foundation.

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

## Appendix A    Comparison with explicitly error-modulated plasticity rules

Normative theories of functional learning in neuronal circuits of the brain commonly derive biologically plausible solutions for the credit assignment problem that approximate BP. Like BP, these models result in explicitly error-modulated update rules for synaptic weights of the general form

$$\Delta w_{ij} \propto \text{pre}_j \times g(\text{post})_i \times e_i \tag{12}$$

where $\text{pre}_j$ is presynaptic neuronal activity, $g(\text{post})_i$ is a nonlinear function $g(.)$ of postsynaptic neuronal activity and $e_i$ is a neuron-specific error signal that determines the sign of plasticity. In this section, we showcase how error-modulated plasticity rules drive neuronal activity to a stable fixed point and discuss their predictions for *in vitro* electrophysiology experiments (Supplementary Fig. S1).

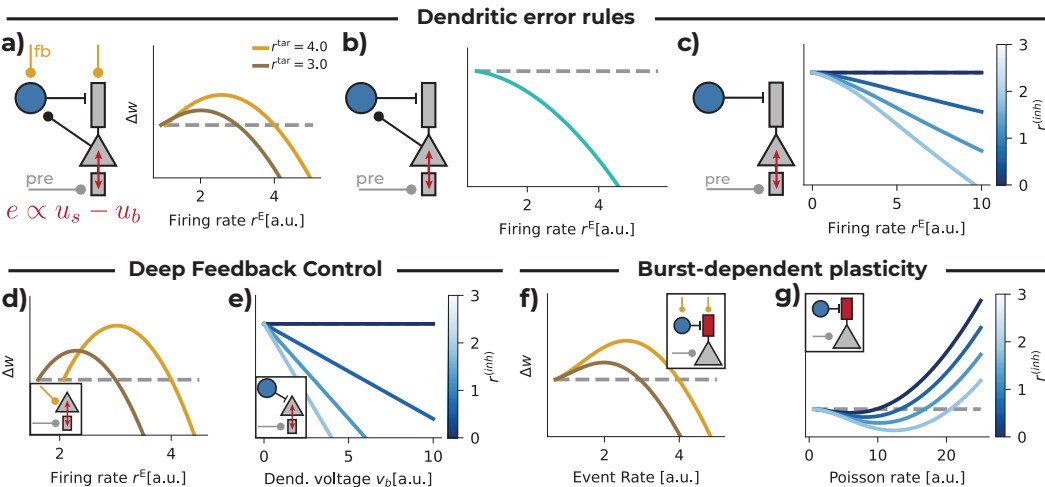

Figure S1:  Simulated *in vitro* patch clamp experiments with different learning rules proposed for bio-plausible credit assignment. **(a)** Dendritic error rules [1] encode the error (red) in the difference between somatic and basal dendritic voltage. When top-down feedback nudges neuronal activity towards a target, this results in error-modulated learning with a stable fixed point at $r^{\text{E}} = r^{\text{tar}}$. **(b)** In the absence of a closed-loop feedback circuit, dendritic error rules postulate that the dendritic potential is canceled out by recurrent inhibition, which would result in weight changes limited to LTD. **(c)** When isolating a single neuron without recurrent connections from excitatory to inhibitory neurons, no weight change can take place except by experimentally injecting currents into dendritic compartments. Increasing inhibitory activity experimentally should thus cause LTD. **(d)** DFC models [2, 3] use a dendritic error learning rule that compares dendritic (only feed-forward) to somatic (including feedback) activity. In the presence of a top-down controller, this leads to error-modulated learning. **(e)** Similar to (c), in the absence of top-down feedback, the sign of plasticity is determined by externally injected current into the somatic compartment. **(f)** In models using burst-dependent plasticity [4, 5], the error (red) is encoded in the deviation of a baseline apical membrane potential, which determines the burst probability of the neuronal output. In a closed-loop circuit, this leads to error-modulated learning. **(g)** If firing rates are Poisson distributed and thus burst probability increases as a function of firing rate, such models can resemble plasticity with a postsynaptic threshold. In this adapted model, increasing inhibitory currents into the apical compartment can bias the learning rate towards LTD.

**Dendritic error rules.**    Dendritic error rules employ neuron models with multiple segregated compartments to encode a neuron-specific error signal $e_i$ in the difference between neuronal activity of different compartments [1, 6]. Here, we implement the model described by Sacramento et al. [1], in which the error is encoded in the difference between a somatic voltage $u_{\text{S}}$ and a basal dendritic voltage $u_{\text{B}}$. For simplicity, we consider a single pyramidal neuron receiving presynaptic input $x$ weighted by a feed-forward weight $w$. Top-down feedback provides a current to an apical dendritic compartment $u_{\text{A}}$ that is canceled out by recurrent inhibition, leaving an error signal in the apical dendrite that in turn influences the somatic voltage (Supplementary Fig. S1a). The error signal used

for the bottom-up weight update is then decoded from the difference between the output of the neuron, $r = \phi(u_\mathrm{S})$, and the output that would be observed if the apical dendrite potential (i.e. the top-down error) was zero. Specifically, the weight change at a synapse with presynaptic input $x$ is given by

$$\Delta w^{\text{Dendr. Error}} = \eta \left( \phi \left( u_\mathrm{S} \right) - \phi \left( \hat{u}_\mathrm{B} \right) \right) x \tag{13}$$

where $\hat{u}_b$ is a function of the dendritic potential that takes into account dendritic attenuation parameters of the model, $\phi(.)$ is a nonlinear activation function and $\eta$ is a learning rate (see Sacramento et al. [1] for a detailed description of the model dynamics).

When this microcircuit is intact, top-down feedback is proportional to a target firing rate $r^{\text{tar}}$ and the neuron-specific error $e_i$ determining the sign of plasticity is proportional to the output error $r^{\text{tar}} - r$. Consequently, the synaptic weight will increase when the neuronal firing rate is below the target rate and vice versa (Supplementary Fig. S1a).

In contrast, when the apical dendrite does not receive top-down inputs, for example when top-down afferents are severed in an *in vitro* slice preparation, the plasticity rule in Eq. (13) cannot decode an error. In this setting, recurrent inhibition would still try to cancel out an expected top-down input to the apical compartment, leading to a negative apical potential and in turn to LTD (Supplementary Fig. S1b). Note that in other implementations of dendritic error rules, that do not employ a recurrent inhibitory circuit [6, 7], $\Delta w = 0$ in this setting.

If the local inhibitory microcircuit is also impaired, for example by blocking recurrent connections from excitatory neurons to inhibitory neurons, the apical voltage is independent of presynaptic stimulation. In this isolated neuron setting, which arguably resembles most experimental conditions for *in vitro* electrophysiology, dendritic error rules would predict no weight change, except if current is injected into the apical compartment experimentally, in which case the weight change would be proportional to the injected current (Supplementary Fig. S1c)

**Deep Feedback Control.**     The DFC framework put forward by Meulemans et al. [2, 3] also makes use of a dendritic error rule, but considers a simpler microcircuit without recurrent inhibition. The weight change in DFC is proportional to the difference between a neuron's bottom-up inputs, encoded in a basal dendritic potential $u_\mathrm{B}$, and its somatic potential $u_\mathrm{S}$. The somatic potential integrates both bottom-up inputs and a top-down control signal that moves the neuronal firing rate towards the target $r^{\text{tar}}$. The DFC update rule is defined as

$$\begin{aligned} \Delta w^{\text{DFC}} &= \eta \left( u_\mathrm{S} - u_\mathrm{B} \right) x \tag{14} \\ &= \eta \left( u_\mathrm{S} - wx \right) x \quad . \tag{15} \end{aligned}$$

When the top-down control loop is intact, the DFC learning rule resembles the learning dynamics of dendritic error rules and creates a stable fixed point at $r = r^{\text{tar}}$ (Supplementary Fig. S1d). As was the case in dendritic error rules, when the top-down control signal is impaired, the sign and magnitude of the weight change are proportional to externally injected currents (Supplementary Fig. S1e).

**Burst-dependent plasticity.**     Payeur et al. [4] suggested an alternative mechanism to leverage segregated neuronal compartments for error-driven plasticity. As in dendritic error rules, burst-dependent plasticity relies on top-down feedback connections to an apical dendritic compartment. However, in contrast to the previously described learning rules, burst-dependent plasticity does not rely on compute $e_i$ using differences in neuronal activity between compartments. Instead, burst-dependent plasticity exploits the unique spiking behavior of cortical layer 5 pyramidal neurons [8], in which inputs to the apical dendrite determine whether bottom-up inputs to the neuron result in a single output spike ("event") or several output spikes fired in quick succession ("burst"). This approach allows the neuron to multiplex two streams of information in its firing rate: bottom-up forward signals are encoded in the event rate $r_{\text{event}}$ while top-down errors are encoded in the burst probability $p$. To illustrate the dynamics arising from burst-dependent plasticity, we consider the rate-based version of the model proposed by Greedy et al. [5], for which the weight change is defined as

$$\Delta w^{\text{Burst-dependent}} = \eta \, r_{\text{event}} \left( p - p_b \right) x \tag{16}$$

where $p_b$ is a parameter denoting the baseline burst probability. The sign of plasticity is thus determined by deviations from the baseline burst probability (see Greedy et al. [5] for details on model dynamics).

In the intact, closed-loop microcircuit, top-down feedback communicates an error signal to the apical compartment that in- or decreases the burst probability depending on whether the neuronal output $r_{\text{event}}$ is smaller or larger than the target rate $r^{\text{tar}}$. Similar to other error-modulated plasticity rules, this creates a stable fixed point for plasticity at $r_{\text{event}} = r^{\text{tar}}$ (Supplementary Fig. S1f).

Because the baseline burst probability $p_b$ acts like a postsynaptic plasticity threshold, burst-dependent plasticity can resemble some aspects of phenomenological Hebbian plasticity rules. Specifically, when synaptic inputs to the neuron follow a Poisson distribution over time, its output will also be Poisson distributed and the burst probability is not solely determined by top-down apical inputs. To illustrate this scenario, we consider an isolated neuron without top-down feedback to its apical dendrite. Given Poisson-distributed inputs $x$, we can assume that the burst probability of the postsynaptic neuron is determined by the input Poisson rate, i.e. $p = f(x)$. Plotting the weight change as a function of the input Poisson rate while keeping the baseline burst probability $p_b$ fixed results in a learning rule that resembles phenomenological Hebbian plasticity with a postsynaptic threshold (Supplementary Fig. S1g).

## Appendix B    Analysis of the learning rule for dis-inhibitory control

Let's first recall the learning rule expressed in Eq. (9), where local error is encoded in the difference between the excitatory firing rate and inhibitory membrane potential. To enhance clarity, we reformulate the weight change for a singular synapse, $w_{ij}$, established between presynaptic excitatory neuron $j$ and postsynaptic excitatory neuron $i$:

$$\Delta w_{ij} = \phi'\left(u_i^{\text{E}}\right)\left(r_i^{\text{E}} - u_i^{\text{I}}\right) r_j^{\text{E}}. \tag{17}$$

where we assume all quantities are evaluated at equilibrium state. Here, $r_j^E$ and $r_i^E$ represent the pre- and post-synaptic firing rates respectively. $u_i^E$ symbolizes the postsynaptic membrane potential, whereas $u_i^I$ denotes the membrane potential of the inhibitory neuron linked to the postsynaptic cell. This learning rule is non-local since the postsynaptic neuron doesn't possess direct access to the membrane potential of an inhibitory neuron $u_i^I$.

To navigate around this limitation, we introduced a linear approximation of the inverse inhibitory activation function in Eq. (11), yielding a local learning rule of the generic form:

$$\Delta w_{ij} = \phi'\left(u_i^{\text{E}}\right)\left(r_i^{\text{E}} - \theta - \delta r_i^{\text{I}}\right) r_j^{\text{E}}. \tag{18}$$

The simplified microcircuit model we studied here assumes a particular form of one-to-one connectivity in which each excitatory neuron only receives input from one inhibitory neuron and vice-versa. Conceptually, $r_i^I$ can thus be interpreted as the inhibitory current received by the postsynaptic neuron.

### B.1    Relationship between $\mathcal{L}$ and $\mathcal{H}$

Following [3], it is straight forward to show that a reduction in $\mathcal{H}$ leads to a reduction in $\mathcal{L}$ by observing that, if $\mathcal{H}$ is minimized to zero, $\mathcal{L}$ is also minimized to zero. This is simple to show by observing that $\mathcal{H} = 0$ only occurs when $\mathcal{L} = 0$ (cf. Eq. (4)), i.e. when the uncontrolled network output equals the target. Formally, for any given loss that evaluates to zero if $\mathbf{r}_L = \tilde{\mathbf{r}}_L^{\text{tar}}$, i.e. $\mathcal{L}(\mathbf{x}, \mathbf{x}) = 0$, we have

$$\mathcal{H} = \sum_n \|Q\check{\mathbf{c}}(n)\|_2^2 = 0 \iff \sum_n \mathcal{L}\left(\tilde{\mathbf{r}}_L(n), \mathbf{r}_L^{\text{true}}(n)\right) = 0 \tag{19}$$

where $\tilde{\mathbf{r}}_L$ denotes the output of the network in the absence of a controller. However, in contrast to the networks considered in [3], the dis-inhibitory controller considered in this work is not guaranteed to minimize the surrogate loss $\mathcal{H}$, and thus $\mathcal{L}$ to zero.

### B.2    Linear approximation of the inverse function

The learning dynamics described by the learning rule in Eq. (11) depend implicitly on the chosen nonlinear activation function $r = \phi(u)$, which dictates the error induced by the linear approximation. For all simulations in this work, we use a soft rectifying nonlinearity as neuronal activation function:

$$\phi(u) = \beta \log(1 + \exp(u - \gamma)) \tag{20}$$

which is parameterized through its scale $\beta$ and shift $\gamma$. Additionally, the learning dynamics are contingent on the choice of linearization parameters $\theta$ and $\delta$ used to approximate the inverse activation function,

$$\phi^{-1}(r) = \gamma + \log\left(\exp\left(\frac{r}{\beta}\right) - 1\right) \approx \theta + \delta r. \tag{21}$$

Fig. S2 depicts functionally and phenomenologically different learning dynamics in the same three settings we already investigated in Section 4, that can be obtained by changing the linearization parameters.

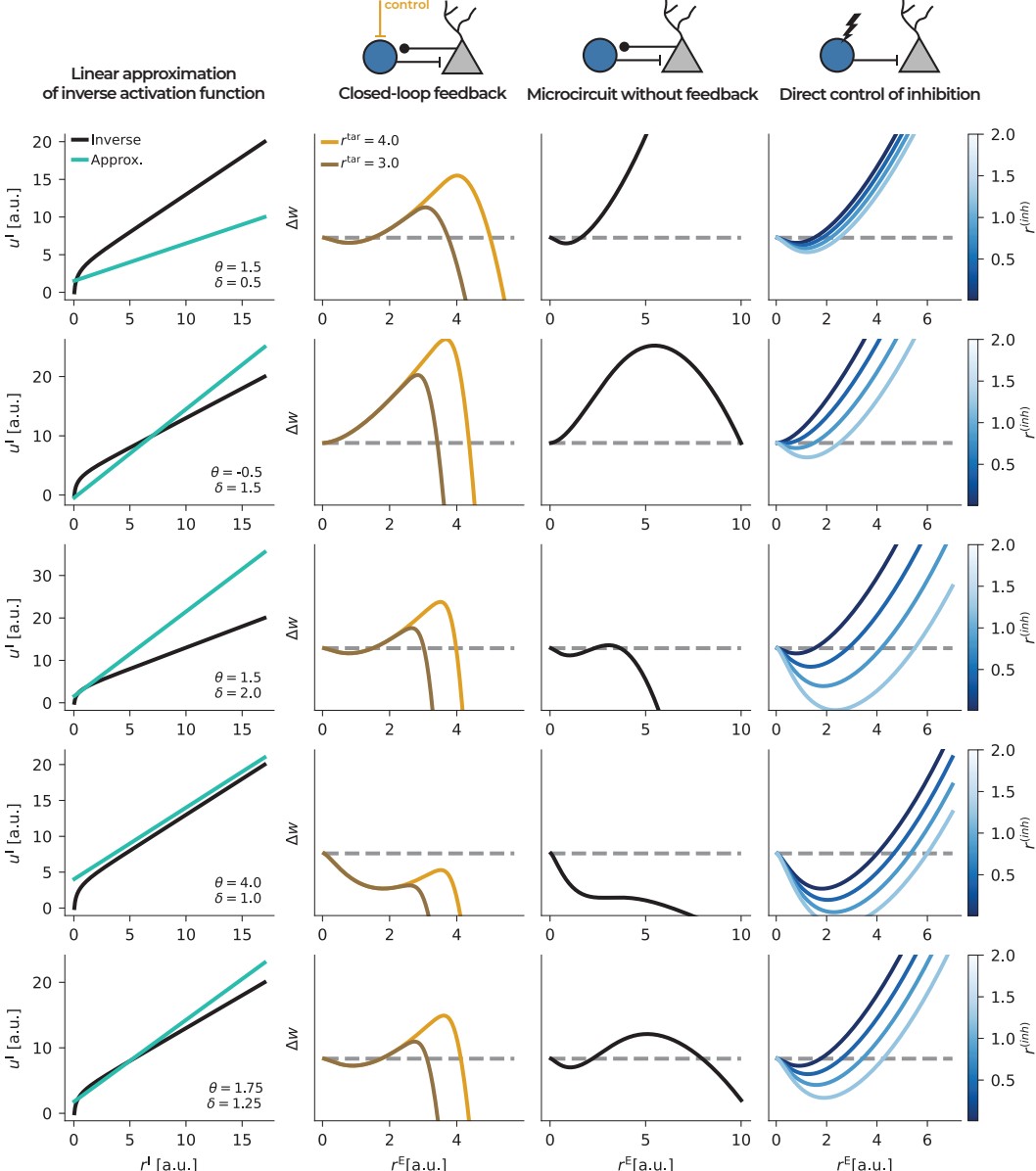

Figure S2: Effect of varying linearization parameters $\theta$ and $\delta$ on the learning rule in different microcircuit conditions. The first column depicts the inverse activation function Eq. (21) and several linear approximations thereof, parameterized through the intercept $\theta$ and slope $\delta$. The second column depicts the dynamics of the resulting learning rule in a microcircuit under the influence of a top-down disinhibitory controller. The third column depicts the learning dynamics in the absence of top-down control. The last column depicts the learning dynamics under simulated experimental conditions, with direct control over the inhibitory current.

For the training of the deep neural networks discussed in Section 5, we adopted a practical approach and linearize the inverse function using a first-order Taylor expansion. This method effectively reduces the linear approximation to a single hyperparameter, the linearization point $\tilde{r}$. Therefore, we approximate the inverse activation function as follows:

$$\phi^{-1}(r) = f(r) \approx f(\tilde{r}) + f'(\tilde{r})(r - \tilde{r}), \tag{22}$$

where $f'(\tilde{r})$ denotes the derivative of the inverse activation function evaluated at the linearization point $\tilde{r}$. From this linearization, we can extract the parameters $\theta$ and $\delta$ in relation to $\tilde{r}$:

$$\theta = f(\tilde{r}) - \tilde{r}f'(\tilde{r}) \qquad \delta = f'(\tilde{r}). \tag{23}$$

Denoting $\tilde{u} = f(\tilde{r}) = \phi^{-1}(\tilde{r})$, we can use the inverse function theorem to formulate the relationship between the linearization point $\tilde{r}$ and the general linearization parameters, $\theta$ and $\delta$, in a more interpretable fashion:

$$\theta = \tilde{u} - \frac{\tilde{r}}{\phi'(\tilde{u})} \qquad \delta = \frac{1}{\phi'(\tilde{u})}. \tag{24}$$

## Appendix C    Simulation details

All numerical simulations were implemented using Python 3.8.10 and were executed on NVIDIA Quadro RTX 5000 GPUs. The software stack includes Jax [9], Flax [10], and Diffrax [11]. Datasets were obtained from the TensorFlow datasets library [12]. The code to reproduce our results is publicly available at https://github.com/fmi-basel/disinhibitory-control.

### C.1    Single-synapse experiments

For the investigation of learning dynamics at a single synapse (Section 4), we simulated a single microcircuit unit. The microcircuit unit was driven by increasing the strength of afferent input into the excitatory neuron. To capture the input-output curve of neurons in the absence of any background inputs, we parameterized the neuronal activation function Eq. (20) with $\beta = 1.0$ and $\gamma = 3.0$ for both excitatory and inhibitory neurons. For each input strength, the microcircuit was simulated for 600 ms to ensure that neuronal dynamics settled to an equilibrium. Synaptic weight updates where computed at the equilibrium state. Numerical integration was performed using the forward Euler method with a time step of 1 ms.

### C.2    Student-teacher task

**Inputs.**    For the student-teacher learning task (Section 5), we generated 30 sine wave inputs with random frequency, amplitude and phase shift. Frequencies were randomly drawn from a uniform distribution $\mathcal{U}(0.1\text{Hz}, 2.0\text{Hz})$. Similarly, sine amplitudes were randomly drawn from the distribution $\mathcal{U}(0.1, 1.0)$ and phase shifts were drawn from the distribution $\mathcal{U}(0, \pi)$. Sine waves were continuously generated and used as input to both the student and teacher network during training. Evaluation for panel B in Fig. 4 before and after training was performed on a 5-second window of the training data.

**Feedback weights.**    Feedback weights $\mathbf{Q}_i$ were initialized proportional to the normalized network Jacobian at the beginning of training:

$$-\mathbf{Q}_i(0) = \alpha \frac{\mathbf{J}_i(0)}{\|\mathbf{J}_i(0)\|_F} \tag{25}$$

where we normalized the Jacobian by dividing it by its Frobenius norm. The parameters $\alpha$ controls the norm of the effective Jacobian, and thus the magnitude of the feedback weights. We found that normalizing the Jacobian is not necessary for good training performance, but prevents exorbitantly large values in the feedback weights and can sometimes speed up numerical integration of the neuronal dynamics and aid the stability of the dynamical system.

As the forward weights $\mathbf{W}_i$ change over the course of learning, the network Jacobian of the student network relevant for computing suitable feedback weights changes as well. To account for the learning-induced changes in the network Jacobian, we make the feedback weights $\mathbf{Q}_i$ part of the

network dynamics and continuously nudge them towards the normalized network Jacobian with a slow time constant $\tau_Q$:

$$\tau_Q \frac{d\mathbf{Q}_i}{dt} = -\mathbf{Q}_i(t) + \alpha \frac{-\mathbf{J}_i(t)}{\|-\mathbf{J}_i(t)\|_F} \tag{26}$$

As a result, feedback weights are dynamically adjusted to retain good control performance. We chose $\tau_Q$ to be faster than the plasticity time constant $\tau_W$ and much slower than changes in the input $x(t)$. As a result, the feedback weights reflect the average network Jacobian over time, but adapt to changes in the neuronal dynamics caused by feed-forward plasticity. Meulemans et al. [2, 3] have demonstrated that feedback weights reflecting the average Jacobian can also be learned using a simple, fully local anti-Hebbian learning rule based on noise correlations between neuronal activity and top-down control.

**Training.** Both student and teacher networks were randomly initialized using Kaiming initialization and hyperparameters relating to numerical integration and neuronal dynamics were kept identical for student and teacher networks (see Table S1). Student networks were trained continuously for a total of 5 minutes, split into 60 periods each lasting 5 seconds. For each 5-second window, we first computed the output of the teacher network to obtain a target for the top-down controller. We then computed the output of the student network in an open-loop setting without top-down control to calculate the MSE loss $\mathcal{L}$. Finally, we simulated the student network with active top-down control and continuously monitored the least-control loss $\mathcal{H}$. Numerical integration was performed using Tsitouras' 5th order explicit Runge Kutta method [13] with an adaptive step size.

Table S1: Hyperparameters used for training networks on the student-teacher task (Fig. 4).

| Parameter | Description | Value |
|---|---|---|
| $\tau_E$ | Exc. membrane time constant | 20 ms |
| $\beta_E$ | Exc. activation function scale | 5.0 |
| $\gamma_E$ | Exc. activation function shift | 3.0 |
| $\tau_I$ | Inh. membrane time constant | 5 ms |
| $\beta_E$ | Inh. activation function scale | 1.0 |
| $\gamma_E$ | Inh. activation function shift | 3.0 |
| $\tau_c$ | Controller time constant | 100 ms |
| $k_p$ | Proportional control strength | 30.0 |
| $k_i$ | Integral control strength | 15.0 |
| $\tau_W$ | Weight update time constant | 60 s |
| $\tau_Q$ | Feedback weight update time constant | 30 s |
| $\tilde{r}$ | Linearization point for learning rule | 0.5 |
| $\alpha$ | Jacobian norm. constant | 7.5 |

### C.3 Computer vision benchmarks

**Data preprocessing and network architecture.** MNIST and Fashion-MNIST images were rescaled to values between 0 and 1 and flattened before being used as inputs to the networks. Each hidden layer was composed of 256 excitatory-inhibitory microcircuit units for networks trained with disinhibitory control or 256 neurons for networks trained with BP.

**Classification with Softmax and cross-entropy loss.** Conventionally, classification tasks are solved using a Softmax layer as output of the network. The Softmax layer is trained to match a one-hot encoded label $\mathbf{r}_L^{\text{tar}}$. As pointed out by Meulemans et al. [3], because the Softmax layer output only matches the one-hot vector when it has an infinite input, this can lead to problems when feedback control is strong. Thus, we use soft targets where the value of $\mathbf{r}_L^{\text{tar}}$ is 0.99 for the correct target and $0.01/n_L-1$ for all other entries. Following [3], we use a linear output layer and absorb the Softmax operation into the cross-entropy loss function:

$$\mathcal{L}^{\text{classification}} = -\sum_n \mathbf{r}_L^{\text{tar}}(n) \log\left(\text{Softmax}\left(\mathbf{r}_L(n)\right)\right) \tag{27}$$

Our dis-inhibitory controller and the associated learning rules require a nonlinear activation function and recurrent inhibition, which could interfere with the Softmax in the loss function. Thus, we

Table S2: Hyperparameters used for training networks on computer vision benchmarks

| Parameter | Description | Value |
|-----------|-------------|-------|
| $\tau_E$ | Exc. membrane time constant | 20 ms |
| $\tau_I$ | Inh. membrane time constant | 5 ms |
| $\beta$ | Activation function scale | 1.0 |
| $\gamma$ | Activation function shift | 0.0 |
| $\tau_c$ | Controller time constant | 100 ms |
| $k_p$ | Proportional control strength | 0.2 |
| $k_i$ | Integral control strength | 0.4 |
| $\alpha$ | Jacobian norm. constant | 1.0 |

implemented the readout layer for classification tasks as a linear layer without excitatory-inhibitory units:

$$\tau_E \frac{d}{dt}\mathbf{r}_L = -\mathbf{r}_L(t) + \mathbf{W}_L \mathbf{r}_{L-1}^E(t) + \mathbf{Q}_L \mathbf{c}(t) \tag{28}$$

with $\mathbf{Q}_L = I$, and performed weight updates according to the deep feedback control learning rule [3].

**Hyperparameter optimization.** Optimal values for the controller parameters $k_p$ and $k_i$ were obtained using the Optuna package [14]. Optimization was performed using a Tree-structured Parzen Estimator [15] with 25 starts. Specifically, hyperparameters were chosen to minimize the validation loss of a 3-layer network after 20 training epochs using the exact inverse learning rule on the MNIST dataset.

**Training.** Network weights were initialized using Xavier initialization and trained for 50 epochs with a batchsize of 100. For every input sample, we first let the network settle to an equilibrium in the absence of feedback control. This uncontrolled equilibrium state was used to calculate the feedback weights $\mathbf{Q}$ and used as initial state for the second convergence with top-down control. Numerical integration was performed using Tsitouras' 5th order explicit Runge-Kutta method [13] with an adaptive step size for a maximum duration of 2 seconds or until equilibrium was reached, defined by an absolute tolerance value of $10^{-6}$. Synaptic weight updates were performed at the controlled equilibrium state and averaged across each minibatch. For all experiments, we used the ADAM optimizer with with learning rate 0.001.

**Feedback weights.** Feedback weights were calculated proportional to the normalized network Jacobian:

$$-\mathbf{Q}_i = \alpha \frac{\tilde{\mathbf{J}}_i}{\|\tilde{\mathbf{J}}_i\|_F} \tag{29}$$

where $\tilde{\mathbf{J}}_i$ is the Jacobian obtained at the uncontrolled equilibrium state (cf. Eq. (7)). Because this calculation results in input-dependent feedback weights, which are biologically implausible, we also consider the case in which feedback weights reflect the average Jacobian. As demonstrated in Meulemans et al. [2, 3], such feedback connections could be learned locally making use of noise correlations between neuronal compartments receiving top-down feedback and the strength of the control signal. Here, we instead take a practical approach and take the average of the Jacobian across each minibatch:

$$-\mathbf{Q}_i^{(\text{Avg } \mathbf{J})} = \alpha \frac{\tilde{\mathbf{J}}_i^{\text{Avg}}}{\|\tilde{\mathbf{J}}_i^{\text{Avg}}\|_F} \tag{30}$$

$$\tilde{\mathbf{J}}_i^{\text{Avg}} = \frac{1}{N}\sum_{n=1}^{N}\tilde{\mathbf{J}}_i \tag{31}$$

Table S3: Validation accuracy in % for networks trained with BP or dis-inhibitory feedback control. Reported values are mean +/- stdev ($n = 10$).

| | MNIST | | Fashion-MNIST | |
|---|---|---|---|---|
| | 1 hidden layer | 3 hidden layers | 1 hidden layer | 3 hidden layers |
| BP | $98.0 \pm 0.24$ | $98.2 \pm 0.14$ | $90.1 \pm 0.29$ | $90.3 \pm 0.34$ |
| Exact inverse | $97.5 \pm 0.19$ | $97.5 \pm 0.19$ | $90.0 \pm 0.24$ | $90.0 \pm 0.38$ |
| Linear threshold | $96.9 \pm 0.16$ | $96.4 \pm 0.18$ | $88.8 \pm 0.25$ | $88.1 \pm 0.25$ |
| Exact inverse (Avg $\mathbf{J}$) | $97.6 \pm 0.15$ | $96.9 \pm 1.6$ | $89.8 \pm 1.39$ | $89.2 \pm 0.48$ |
| Linear threshold (Avg $\mathbf{J}$) | $96.4 \pm 0.15$ | $95.8 \pm 0.21$ | $87.9 \pm 0.36$ | $86.2 \pm 0.52$ |

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
