# OpenReview forum: "Dis-inhibitory neuronal circuits can control the sign of synaptic plasticity"
_NeurIPS.cc/2023/Conference — NeurIPS 2023 poster_

### Official Review · Reviewer_TQSB · 2023-06-14

**Soundness:** 4 excellent
**Presentation:** 4 excellent
**Contribution:** 2 fair
**Rating:** 6
**Confidence:** 5

**Summary:**

The authors extend previous work on deep feedback control (DFC) learning to make the functional form of plasticity a more realistic reflection of what is observed biologically. In particular the authors:
1. Introduce a feedback control mechanism using targeted, neuron-specific inhibition signal that allows for synaptic plasticity that more closely resembles experimentally observed plasticity (e.g. BCM), compared to the error-based learning employed in a wide variety of previous papers.
2. Show that learning with the extended plasticity rule still comes relatively close to backpropagation-level performance on several benchmarks (MNIST, FMNIST).
3. Discuss at length the testable predictions of their extended model.

**Strengths:**

The strengths of the paper are as follows:
1. Reducing an algorithm derived from optimization principles to the point that it can replicate LTD/LTP experiments is quite difficult, and many (but not all) preceding algorithms do not appear to succeed in this regard, relying more on error-based signals that are less clearly related to experimentally observed plasticity phenomena.
2. Even with biophysically motivated modifications, the algorithm still performs quite well on image classification tasks, which is also difficult.
3. The paper is very clearly written and organized.

**Weaknesses:**

The authors themselves identify several key weaknesses, which I will elaborate on below. However, to me the principal weakness is that the contribution is very incremental: previous studies (e.g. Payeur et al. 2021) have demonstrated that related learning algorithms can, in certain regimes, resemble BCM-like learning, and the high performance and 'locality' of the DFC family of algorithms has already been explored extensively in previous studies (e.g. Meulemans et al. 2021 & 2022). Therefore, it seems to me that the key improvement this study demonstrates is that the DFC algorithms can, with some extra tweaks, also resemble this type of Hebbian learning.

Other weaknesses:
1. As the authors note, their current learning scheme involves a controller with access to a highly complex Jacobian. This Jacobian is a function of the fixed-point of the network dynamics, and so as far as I can tell, for every individual stimulus, the network's feedback weights would have to be different in order to match the principled feedback control dynamics. Previous DFC studies demonstrated that it's possible to get away with simpler controllers with fixed weights, but this approximation is not used in this study, and it's not clear why--as is, though the learning rule locally resembles BCM, the feedback signal actually used to achieve high performance on MNIST/FMNIST is essentially almost as complicated than the backpropagation error signal itself.
2. The unrealistic 1-1 mapping between inhibitory and excitatory neurons makes it difficult to pin down where exactly the controller feedback should be expected to be at the level of a cortical microcircuit.

**Questions:**

Could you elaborate on the relationship between this algorithm and the algorithm proposed in Payeur et. al 2021? In particular, that algorithm also replicates BCM-like LTD/LTP phenomena--it seems as though the ability to replicate BCM-like learning dynamics is not unique to the DFC family of models. What are the key differences in testable predictions between this model and Payeur 2021? Or older predictive coding-based models like Urbanczik & Senn 2014?

Is the method employed in this study inherently unique to the DFC family of algorithms, or does it apply equally well to the other algorithms as well? E.g., could similar interneuron-modulated plasticity also be applied to predictive coding-based models?

If the network is driven to a near-optimal performance regime by the controller for every stimulus, and this is occurring in a biological system, would there ever be any observable improvements in performance? Or would the animal be performing instantaneously very well, with the only observable progressive change being a reduction in the energy required for the controller? If this is true, which biological systems could this algorithm adequately model?

In this model, is top-down feedback exclusively isolated to inhibitory neurons? Is this compatible or incompatible with models that propose feedback is also (or exclusively) directed to apical dendrites of pyramidal neurons?

**Limitations:**

There are no obvious negative societal impacts of this work, and the authors very adequately address the limitations of their work.

---

> ### Author Rebuttal · Authors · 2023-08-09
>
> > (...) to me the principal weakness is that the contribution is very incremental: (...)
>
> While our results are comparable to previous work in terms of performance, we argue that our model offers distinct computational and conceptual advantages:
>
> First, the dis-inhibitory controller we use causes the error signal to be encoded in the inhibitory current received by each neuron. This alleviates the need for the neuron to distinguish between its feed-forward inputs and overall output rate.
>
> Second, our work demonstrates that a Hebbian learning rule with a minor tweak can decode top-down dis-inhibitory error signal and thus allow top-down feedback to steer the sign of plasticity. This learning rule resembles phenomenological plasticity models and thus provides a plausible mechanism through which credit signals can influence local plasticity rules. Thus, our work proposes a concrete circuit-level mechanism for error coding through modulation of recurrent inhibition.
>
> >(... this model) involves a controller with access to a highly complex Jacobian. (...) for every individual stimulus, the network's feedback weights would have to be different (...)
>
> We have performed additional simulations in which the feedback weights are proportional to a stationary Jacobian. This actually improved the stability of our model and shows that our model’s performance is robust to more plausible feedback weights. (cf. Section 1 of the general rebuttal and PDF Tables 1-2 for details).
>
> >The unrealistic 1-1 mapping between inhibitory and excitatory neurons makes it difficult to pin down where exactly the controller feedback should be expected (...)
>
> We agree that the one-to-one connectivity in our model is unrealistic. However, this minimal model is sufficient to illustrate the mechanism of dis-inhibitory control. It is yet unclear where top-down control of cortical circuits originates. In any case, we believe that dis-inhibitory interneurons (such as VIP+ cells) could be directly targeted by a controller. Alternatively, top-down feedback could consist of balanced excitatory and inhibitory currents. In this case, VIP+ interneurons could act as a gating mechanism for feedback control.
>
> >Could you elaborate on the relationship between this algorithm and the algorithm proposed in Payeur et. al 2021? (...)
>
> In Payeur et al. (2021), the sign of plasticity is controlled by deviations from a baseline burst probability. Their proposed mechanism is plausible for cortical L5 neurons with a separate apical dendritic compartment spatially segregating top-down and bottom-up inputs. It is not clear how a burst-dependent mechanism would work for cortical L2/3 neurons or other brain areas that lack segregated dendritic compartments and do not show stereotypical burst firing.
>
> >What are the key differences in testable predictions between this model and Payeur 2021? Or (...) Urbanczik & Senn 2014?
>
> The main difference between these three models is which quantity determines the sign of plasticity. Urbanczik & Senn (2014) postulates that the sign of plasticity depends on differences between local dendritic and somatic voltage. Burstprop (Payeur et al. (2021)) postulates that the sign of plasticity depends on the proportion of bursts in the output of the neuron. Our paper postulates that the amount of inhibitory current into the neuron determines the sign of plasticity.
>
> >(...) does (inhibition-modulated plasticity) apply equally well to the other algorithms as well? (...)
>
> Yes! Thanks for asking. Although we derived our model within an adaptive control theory framework with a strong controller (strong nudge), the concept of decoding error signals from changes in inhibitory activity equally applies to models with weak feedback (weak nudge). For weakly nudged networks, our model is directly related to predictive coding models which are usually considered in the weak nudge limit (but see also Song, [...], Bogacz (2023)).
>
> >If the network is driven to a near-optimal performance (...) would there ever be any observable improvements in performance? (...) which biological systems could this algorithm adequately model?
>
> No, we do not expect the animal to perform instantaneously very well. Feedback signals can only be computed when the animal received the feedback for its actions, e.g. a reward.
>
> For simplicity, we considered the idealized regime in which an agent is presented with a completely unambiguous target signal immediately. A behavioral paradigm that comes close to this setting is fear conditioning, where a painful unconditioned stimulus (US) is a strong, unambiguous value target. Learning is expressed as a freezing response to the initially neutral conditioned stimulus (CS). If a control mechanism exists that drives a network initially responsive to the CS to respond to a target value given by the US, neurons that are part of the top-down controller should reduce their response to the US proportionally to the learned response to the CS. This has been observed for VIP+ interneurons in the amygdala (Krabbe et al. (2019)), corroborating a role for dis-inhibitory circuits in learning.
>
> >(...) is top-down feedback exclusively isolated to inhibitory neurons? Is this compatible (...) with models that propose feedback (...) to apical dendrites of pyramidal neurons?
>
> In our model, feedback is exclusive to a specific class of inhibitory neurons and affects excitatory neurons indirectly through dis-inhibition. There are presumably different top-down signals in the brain that may originate from different brain areas and preferentially target different cell types. Therefore, we do not consider this an overly restrictive assumption. There is likely other top-down feedback to excitatory neurons, which we did not address here. An alternative implementation to interneuron-mediated feedback could be that dis-inhibitory circuits provide a gating signal for excitatory top-down feedback, which would otherwise be canceled out by inhibition.

---

> > ### Comment · Reviewer_TQSB · 2023-08-10
> > **Response to rebuttal**
> >
> > Thank you for your very detailed feedback. Though I still believe the results in this paper are incremental relative to previous work, and that this is the principal weakness of the paper, you have done a lot to convince me of the rigor of your analyses. Adding these points (especially your additional figures) will certainly increase the quality of the paper. I will maintain my score, but will increase my confidence (score: 6; conf. 5).

---

### Official Review · Reviewer_A5Rt · 2023-06-25

**Soundness:** 3 good
**Presentation:** 3 good
**Contribution:** 3 good
**Rating:** 7
**Confidence:** 4

**Summary:**

The authors propose a neural plasticity mechanism with a key role for disinhibition. In particular, they apply a deep feedback control framework whereby feedback driven inhibitory neurons mediate changes in the feedforward excitatory connections. The proposed rule is argued to hold desirable properties in that it is local, captures/predicts experimentally observed plasticity, and can guide error-modulated learning.

**Strengths:**

- the paper is generally well written (though there are some typos, see below)
- the proposed plasticity is novel and well theoretically groundeded
- the experimental predictions are well presented and appear feasible to perform


**Weaknesses:**

- I am not fully convinced of the extent of novelty within this work. It seems to me that the authors made only slight alterations to the setup in [1, 2] such that inhibitory neurons are now included in the architecture (instead of + Qc we now have - (- Qc)) . Of course the brain does include inhibitory neurons, so adding them explicitly is arguably one step closer to plausibility, but I think the authors could present a stronger argument by addressing more the functional/computational differences/benefits with this addition compared to the previous models [1,2]. Comparing it to theorem 2 in [1], is the key difference that the interneurons enable the network to avoid the need to store/discriminate between the feedforward (ff) activity and the total activity of the excitatory neurons? If so I think this could be more clearly conveyed

- I think the authors could more in relating their work to other computational works which consider the role of inibitory neurons on plasticity. For example, would [3] make different predictions in Fig 2? Moreover, the authors do not relate their model to [4], which to my knowledge seems to have significant intersection in terms of modeling and possibly predictions.

- though I found the writing in general good, there were still a fair few sloppy errors and places which were unclear to me (see below)

References:
[1] Meulemans et al. 2021, Credit assignment in neural networks through deep feedback control
[2] Meulemans et al. 2022, Minimizing Control for Credit Assignment with Strong Feedback
[3] Sacramento et al. 2018, Dendritic cortical microcircuits approximate the backpropagation algorithm
[4] Greedy et al. 2022, Single-phase deep learning in cortico-cortical networks

**Questions:**

- Fig 1a: what's the dfiference between error and feedback signals?
- Fig 1b: what is f and g? Why are they different?
- Fig 1 caption: eqprop not defined
- line 93: "While these error-modulated learning rules prove functionally useful...they fall short of capturing established properties of
experimentally observed plasticity, such as a postsynaptic activity threshold". Forgive me if I'm being naive: what is meant here postsynaptic activity threshold? It's not clear to me.
- line 99: "However, the model (burstprop) assumes a rigid circuit architecture to decode errors from multiplexed spike trains and thus does not generalize to other neuronal circuits". I don't understand the logic othis sentence: could the authors elaborate?
- line 102: "..necessitate feedback signals to be weak.."; this sentence confused me a bit. It seems to me there's a difference between 1. the strength of feedback signals and 2. whether feedback signals activity influence neuronal activity. For example, in burstprop one might have a very high apical potential (and burst rate) but little/no influence on the event rate
- line 130: define L
- I found the explanation of how optimal feedback weights are found confusing (equation 7). Firstly, J contains L vectors for each u_i, but this is multiplied by only one u_i? Secondly, it seems equation 7 just shows how the change in r relates to any changes in u - this is true in whatever weights are chosen, I don't understand how this is an equation to be 'solved'. Finally, from what I understand in [1] it's true that that if the column space of Q is equal to the row space of J (line 145) then equivalence to Gauss-Newton optimisation is possible, but is this choice of Q necessarily optimal in this case?
- equation 8: use of subindices meaning datapoints here whilst layers elsewhere is confusing
- lines 152-159: it was only clear to me after looking at [2] that minimising the surrogate loss should also minimise the task loss. This should be presented more clearly
- line 156: I don't see the logic of 'as a result' from the previous sentence; doesn't it just follow from equation 5?
- line 167, 169: Eq. not equation
- Fig 2: is the x axis post-synaptic rates? this should be in caption. Same with frequency in 2d
- Fig 2b: inset diagram is unclear; is the black line the true inverse function? I am also curious to see an imperfect approximation where the approximation is above/below the true values at low/high values respectively
- Fig 2 caption: 'resembles experimentally observed plasticity in simulated in vitro condition' - this seems a strong statement when only one paradigm (an isolated neuron) is actually shown
- line 197: "The dis-inhibitory feedback signal creates a stable equilibrium state for the weight dynamics that coincides with the postsynaptic target computed by the feedback controller". I'm sorry, I didn't understand this sentence
- for the student-teacher task in section 5, what is f? is it the output of a randomly intialised teacher network of the same architecture as the student network but without feedback? If without feedback, why does the solution for H in 3c bottom not go to zero? Also, is it necessary to set the feedback weights as the transposed Jacobian. Would it also work if its column space was equal to the row space of J?
- For simulations in image tasks, n=3 seeds seems quite low to me. Could you repeat for a higher number? (like n=10)
- For readability I'd recommend rotating table 2 (perhaps switching rows with columns is necessary)

typos:
line 105. Full stop after references.
equation 8: bold Q
line 203: appendix section 'xyz' (also correct in appendix itself)
line 207: poor grammar
line 244: appendix C and D
Tabel 1 caption: full stop at end
line 284: comma after 'fashion-MNIST'
line 336: has -> have

References:
see refs for weaknesses above

**Limitations:**

Limitations were well addressed by authors

---

> ### Author Rebuttal · Authors · 2023-08-09
>
> >I am not fully convinced of the extent of novelty within this work. (...) is the key difference that the interneurons enable the network to avoid the need to discriminate between the feedforward (ff) activity and the total activity (...)
>
> We realize that we did not sufficiently explain the advantage of a dis-inhibitory feedback circuit. The core *computational* advantage is that encoding the error in inhibitory activity removes the need for a neuron to store a copy of its feed-forward inputs (e.g. Sacramento et al. (2018), Meulemans et al. (2021, 2022)).
>
> Our model also improves upon existing work *conceptually* by demonstrating that a learning rule resembling phenomenological models of Hebbian plasticity can be used to decode the error and thus allow feedback connections to steer the sign of plasticity. This builds a plausible bridge between prior studies on credit assignment and the translation of credit signals into tangible local weight adjustments.
>
> >I think the authors could more in relating their work to other computational works  (...).
>
> We agree that the comparison to existing models of bio-plausible credit assignment was not sufficient. We have repeated the simulated *in vitro* slice experiments for several related models (see Section 2 of our general rebuttal and Fig. 1 in the attached PDF)
>
> >Fig 1a: what's the dfiference between error and feedback signals?
>
> We interpreted error signals as explicit relay of an error while feedback signals influence neuronal activity.
>
> >Fig 1b: what is f and g? Why are they different?
>
> We mean two nonlinear functions of the postsynaptic activity. Phenomenological models usually assume a calcium trace. Models of credit assignment usually use the derivative of postsynaptic activity.
>
> >line 93: (...) what is meant here postsynaptic activity threshold?
>
> Experiments on Hebbian plasticity have found an activity threshold above which long-term depression changes to potentiation for co-activated synapses. In phenomenological models, this threshold is usually expressed in terms of a postsynaptic quantity (calcium / voltage / rate).
>
> For example, Artola et al. (1990) found that the sign of plasticity was determined by postsynaptic voltage. Sjöström et al. (2011) observed that the sign of plasticity in an STDP protocol depended on postsynaptic firing rate. Lim et al. (2015) found evidence for a postsynaptic activity threshold by investigating *in vivo* firing rate distributions.
>
> >line 99: (...) I don't understand the logic othis sentence (...)
>
> We realize this sentence was not clear. What we meant is that burstprop assumes electrically segregated dendritic trees that are limited to L5 pyramidal cortical neurons. It is unclear whether a similar mechanism would work for L2/3 neurons or other brain areas that do not show such a segregation.
>
> >line 102: (...) It seems to me there's a difference between 1. the strength of feedback signals and 2. whether feedback signals activity influence neuronal activity (...)
>
> Thanks for raising this point. We will restructure our introduction section taking into account this categorization.
>
> >I found the explanation of how optimal feedback weights are found confusing (equation 7). (...)
>
> Thank you for pointing out the lack of clarity. Given the changes to the feedback weights we use (see general rebuttal Section 1), we will update the whole paragraph on the choice of feedback weights $Q$.
>
> >(...) if the column space of Q is equal to the row space of J (line 145) then equivalence to Gauss-Newton optimisation is possible, but is this choice of Q necessarily optimal in this case?
>
> Indeed, the $Q$ we use is just one of many feedback weight matrices fulfilling this condition. In the final manuscript, we'll avoid referring to feedback weights as “optimal” to prevent misinterpretation.
>
> >Fig 2: is the x axis post-synaptic rates? (...)
>
> Yes. We will make this clear in the figure caption.
>
> >Fig 2b: (...) is the black line the true inverse function?
>
> Yes. We will edit the figure caption.
>
> >I am also curious to see an imperfect approximation where the approximation is above/below the true values at low/high values respectively
>
> The impact of imperfect approximations is demonstrated in Figure 4 of the Supplementary Material in our original submission. We will make sure to highlight this supplementary Figure more prominently.
>
> >Fig 2 caption: 'resembles experimentally observed plasticity in simulated in vitro condition' - this seems a strong statement (...)
>
> We will revise the sentence to:
> “(...) resembles plasticity observed in single-neuron electrophysiology experiments”
>
> >line 197: (...). I'm sorry, I didn't understand this sentence
>
> Apologies for the confusion. We mean that the top-down controller influences the learning rule so that a stable fixed point exists when the neuron’s output equals the target.
>
> >for the student-teacher task in section 5, what is f? (...)
>
> $f$ is the output of a randomly initialized teacher network with a smaller hidden layer compared to the student. The teacher network is a conventional ANN without inhibitory units or feedback. One possibility the network does not reach exactly 0 error is the time-delay of the control signal in the online learning paradigm.
>
> >Also, is it necessary to set the feedback weights as the transposed Jacobian? (...)
>
> We have reproduced our results using more relaxed assumptions on the feedback weights (see Section 1 of the general rebuttal and Tables 1-2 in the attached PDF).
>
> ### Other suggestions
>
> Thank you for many more helpful suggestions and finding several typos. We will address all typos and implement the following changes in the revised manuscript:
>
> - Define EqProp in Figure 1 caption
> - (line 156) change the phrase to “Thus, following Eq. (5)...”
> - (line 130) define $\mathcal{L}$
> - repeat all simulations n=10 times
> - rotate table 2
> - change the sub-indices in Eq. (8)
> - (lines 152-159) add a section on the relationship between $\mathcal{H}$ and $\mathcal{L}$

---

> > ### Comment · Reviewer_A5Rt · 2023-08-12
> >
> > Thank you to the authors for the detailed and informative response.
> >
> > I'm impressed with the work the authors have done in the rebuttal and my main concerns have been addressed. Overall, I would not call this model a massive leap forward from the Meulemans et al. works, but I think the authors do make sufficiently novel and interesting predictions for neuroscience, a field in which gains are typically incremental. I will upgrade my score to 7.

---

### Official Review · Reviewer_EPnb · 2023-06-30

**Soundness:** 3 good
**Presentation:** 4 excellent
**Contribution:** 3 good
**Rating:** 7
**Confidence:** 3

**Summary:**

This paper uses adaptive control theory to derive plasticity rules for a fairly plasubile model of multi-layer networks in the brain, which are capable of matching the performance of backpropagation without restrictive assumptions such as mirrored or very weak feedback connections. Specifically, each excitatory neuron has a dedicated inhibitory neuron, which is in turn inhibited by the error signal propagated along feedback connections. The plasticity rule for excitatory units is essentially Hebbian but modulated by the inhibitory signal, matching in vitro plasticity experiments. A brief theoretical derivation of the rule is augmented with experimental results on both toy and relevant problems.

**Strengths:**

* This work presents a relatively well-founded model for feedback of error in the brain, which seems to me one of the most biologically-plausible models of backprop to date.
* All of its presentation is clear and should be accessible to readers with a wide variety of related backgrounds.

**Weaknesses:**

* There are some lingering issues of biological plausibility: There are still significant constraints placed on the feedback weights; each excitatory neuron is assumed to have its own inhibitory neuron (which doesn't match the ratios found in the brain); and Dale's law is not enforced.
* The experimental evaluation is a bit terse; I would like to see the performance of this learning rule on a broader range of tasks (especially a non-toy problem with some temporal structure).

**Questions:**

* Line 150: learning is duplicated
* Figure 2(a): The inset mentioned in the caption seems to be missing

**Limitations:**

All of the concerns I had were addressed thoroughly in the discussion section.

---

> ### Author Rebuttal · Authors · 2023-08-09
>
> >There are still significant constraints placed on the feedback weights
>
> We acknowledge that there are constraints on the feedback weight matrix. Specifically, we used the transpose of the network Jacobian to obtain the feedback weights, which caused feedback weights to vary over time. To address these constraints, we have repeated our simulations with feedback weights derived from a stationary Jacobian. In fact, this has improved our results. Please refer to Point 1 in the general rebuttal and the attached PDF for a thorough discussion of these simulations.
>
> > each excitatory neuron is assumed to have its own inhibitory neuron (which doesn't match the ratios found in the brain)
>
> The proposed model indeed relies on (1) biologically unrealistic one-to-one connectivity and (2) ratios between excitatory and inhibitory neurons. Reducing the number of interneurons to reflect neurobiological ratios has complex implications for the dimensionality of the credit signal, which to address is beyond the scope of this paper. However, we have some exciting preliminary data on low-rank feedback signals (see section 3 in our general rebuttal and Figure 2 in the attached PDF) that suggests that these issues can be overcome. We intend to address this question in the future.
>
> >Dale's law is not enforced
>
> Dale’s law is enforced within each layer, where the E-I connections and I-E connections are purely excitatory or inhibitory, respectively. Output connections from one layer to the next are indeed not Dalian and can be both negative and positive. Such between-layer negative connections should be interpreted as being mediated through feed-forward inhibition.
>
> >The experimental evaluation is a bit terse; I would like to see the performance of this learning rule on a broader range of tasks (especially a non-toy problem with some temporal structure).
>
> Thank you for this suggestion. We were not able to perform simulations on additional datasets due to their computational demand yet. We will try to train networks trained with our learning rule on several datasets with temporal structure, such as the Google Speech Commands dataset for keyword spotting and the Heidelberg Digits or TIDIGITS datasets for classification.
>
> > Line 150: learning is duplicated.
>
> > Figure 2(a): The inset mentioned in the caption seems to be missing
>
> Thank you for spotting these typos. We will fix the mistakes in the revised manuscript.

---

> > ### Comment · Reviewer_EPnb · 2023-08-13
> >
> > Thank you for the detailed response, and especially the additional simulations! Although I do maintain that one-to-one connectivity prevents these results from being a true breakthrough in biologically-plausible backprop, I believe this is a good paper that makes tangible progress and stand by my original score.

---

### Official Review · Reviewer_z6yQ · 2023-07-06

**Soundness:** 2 fair
**Presentation:** 3 good
**Contribution:** 2 fair
**Rating:** 5
**Confidence:** 4

**Summary:**

This paper adds recurrent inhibition to each layer of a DNN architecture in order to facilitate a more biologically-plausible form of credit assignment. It shows how this circuit can explain some of the features of plasticity found in vitro and that DNNs with this circuit can learn to perform simple visual tasks.

**Strengths:**

The microcircuit is biologically motivated

The insights provided into how the artificial experimental conditions in plasticity studies lead to specific results is helpful



**Weaknesses:**

The motivation and innovation was not entirely clear to me. The background is focused on how credit assignment requires distant neurons to interact, yet the problem of how information reaches each layer in the network is not what is tackled here. There is also discussion of the microcircuit "decoding" the credit signal, but it seems like the credit signal is fairly directly given to the layer, and so the microcircuit is more of a relay than a decoder.

One of the main results seems to be that a linear approximation to the inverse activation function can make the weight update rule slightly more biologically realistic and still works fairly well. This is fine, though not a very impactful result.

I found some of the results descriptions confusing (see below)

**Questions:**

Why is there no weight matrix for the recurrent connections?

Can the authors elaborate on how their formulation supports stability? For example, I didn't fully understand this claim "On the other hand, our model suggests that a rapid compensatory mechanism could be implemented as a combination of recurrent inhibitory microcircuits and a linear inhibitory threshold in the synaptic plasticity rule"

"Experiments on excitatory plasticity in vitro are commonly performed in the presence of Tetrodotoxin, a sodium channel blocker, to minimize interference of inhibitory activity" Doesn't TTX block all activity, not just inhibitory?

"Specifically, we block recurrent inhibition " From the diagram it looks like you block recurrent excitation, not inhibition (There would be no point in varying the activity of the inhibitory neuron if its outward connections were blocked).

Is "in the absence of any inhibitory input" supposed to say "absence of input to the inhibitory cells" ?

It's not clear to me why the "postsynaptic" term in equation 9 drops out when the E->I connection is dropped

"we manually set the feedback weights to the transposed network Jacobian at each time step" does this mean that the feedback is different at each timestep? That is not very biologically-plausible.

**Limitations:**

Already mentioned above

---

> ### Author Rebuttal · Authors · 2023-08-09
>
> > The motivation and innovation was not entirely clear to me. (...)
>
> We realize that we did not explain the paper's motivation clearly. What our article contributes is a plausible mechanism consistent with experiments of how neuronal circuits can translate credit signals into weight changes. We do not propose a new strategy on how these credit signals are propagated. Thus our article is more about how feedback afferents can control the sign of plasticity at individual neurons.
>
> We will rewrite the relevant parts in the introduction and discussion sections and propose a new title: “Dis-inhibitory neuronal circuits can control the sign of plasticity”, inspired by Richards & Lillicrap (2018).
>
> > (...) it seems like the credit signal is fairly directly given to the layer, and so the microcircuit is more of a relay than a decoder.
>
> Yes and no. We agree that the feedback is given directly to the layer. But for a neuron to use this feedback for plasticity it needs to estimate the error signal from the magnitude of inhibitory current. Specifically, it needs to correct for its own contribution to the received recurrent inhibition. In that sense, we do think the term “decode” for both the learning rule and the circuit is justified. However, we will work this point out more clearly in our Discussion.
>
> >(...) a linear approximation to the inverse activation function can make the weight update rule slightly more biologically realistic and still works fairly well. This is fine, though not a very impactful result.
>
> We respectfully disagree with this assessment. We think it is impactful because it provides a missing piece of the puzzle of how existing works on credit assignment could plausibly translate credit signals into local weight changes. Most previous models did not link their work to phenomenological learning rules observed in neurobiology.
>
> >Why is there no weight matrix for the recurrent connections?
>
> Our model adopts a one-to-one connectivity structure between excitatory and inhibitory neurons. Although biologically implausible, this minimal architecture is sufficient to demonstrate the central notion of our article: credit signals could be relayed through dis-inhibitory afferents, encoded within inhibitory currents and subsequently decoded using Hebbian plasticity mechanisms.
>
> Other prominent models, such as Sacramento et al. (2018), have employed similar 1-1 connectivity paradigms, underlining their utility in specific contexts. While we have some promising preliminary results on more realistic E-I connectivity patterns (see Section 3 of the general rebuttal and Fig. 2 of the attached PDF), a detailed study of this topic goes beyond the scope of this article.
>
> >Can the authors elaborate on how their formulation supports stability? (...)"
>
> In traditional Hebbian plasticity rules, runaway LTP can lead to unstable dynamics. As highlighted by theoretical studies (e.g, Zenke & Gerstner 2017, Yger & Gilson 2015), plasticity should be accompanied by a rapid compensatory mechanism, i.e. the learning curve should dip down into the LTD region at high firing rates or synaptic weights, to counterbalance a positive feedback loop. However, there is inconclusive evidence for such a mechanism from pair-based-recordings. Interestingly, the model presented in this article shows the desired behavior, but it arises as a circuit phenomenon (See Figure 2a-b in the Manuscript) from an interplay between two features:
> - Recurrent Connectivity: Increases in excitatory firing rate lead to increases in inhibitory current due to recurrent connectivity.
> - Linear error decoding: Without a top-down error signal to decode, the linear model decoding the error from inhibitory activity can overestimate the control signal, causing LTD. The linearization parameters play a key role in the stability of the learning rule, as illustrated in Fig. 4 of the Supplementary Material.
>
> >"Experiments (...) are commonly performed in the presence of Tetrodotoxin (...) to minimize interference of inhibitory activity" Doesn't TTX block all activity, not just inhibitory?
>
> Thank you for pointing this out. TTX is commonly used in glutamate uncaging experiments to reduce interference from background activity. In paired patch-clamp recordings, experimenters instead often apply GABA antagonists, which specifically block inhibitory currents, or discard recordings in which background activity was present, in order to isolate the synapse under investigation.
>
> We will change the sentence to:
> “Experiments on excitatory plasticity are commonly performed under conditions designed to minimize the interference of inhibitory activity, for example by applying GABA antagonists.
>
> >(...) From the diagram it looks like you block recurrent excitation, not inhibition (...).
>
> We will change the sentence to:
> "(...) specifically, we block recurrent connections from excitatory to inhibitory neurons, so that inhibitory activity is independent of excitatory activity (...)"
>
> >Is "in the absence of any inhibitory input" supposed to say "absence of input to the inhibitory cells" ?
>
> We will replace this phrase with “In the absence of any inhibitory activity, i.e. $r^I = 0$, (...) ”.
>
> >It's not clear to me why the "postsynaptic" term in equation 9 drops out when the E->I connection is dropped
>
> Thank you for spotting this error. When the inhibitory firing rate equals zero, the learning rule reduces to $\Delta w \propto r_{\text{pre}} \left(r_{\text{post}} - \theta \right) \phi’(r_{\text{post}})$.
>
> >"we manually set the feedback weights to the transposed network Jacobian at each time step" does this mean that the feedback is different at each timestep? (...).
>
> Thanks for raising this point. Indeed, in our initial experiments, the feedback weights were dynamically adapted at each timestep based on the transposed network Jacobian. We have conducted additional simulations using static feedback weights (see Section 1 in the general rebuttal and Tables 1&2 in the attached PDF).

---

> > ### Comment · Reviewer_z6yQ · 2023-08-14
> >
> > I appreciate the clarifications and corrections from the authors. The intended impact of the work is now clearer. I will increase my score by 1.

---

### Author Rebuttal · Authors · 2023-08-09

We’d like to thank all reviewers for their helpful and constructive comments on our manuscript.  We have responded to each of you individually in our rebuttals below. Based on their collective feedback, we have performed additional simulations which we think strengthen the manuscript. We discuss their  outcomes in the following and we include new tables and figures in the attached PDF document.

### 1. Inhibitory control works with a simpler controller

Several reviewers pointed out that our feedback weights were implausible because they changed at every timestep, i.e. $Q_i(t) = \frac{\partial r_L}{\partial u^I_i(t)}$. We agree that this may look like too much of a constraint.
To address this, we derived the feedback weights from the network Jacobian at the uncontrolled (i.e. $c=0$) steady state. Specifically, we set $Q_i$ proportional to $\frac{\partial r_L}{\partial \tilde u^I_i}$ where $\tilde u^I_i$ are the inhibitory membrane potentials of layer $i$ at the uncontrolled steady-state. Thus, the feedback weights are static within a single trial. We reproduced  all our simulations on computer vision benchmarks in this setting and summarized the results in Table 1 of the attached PDF. All our results still hold in this setting. In fact, using these stationary feedback weights sped up our simulations due to reducing computational demand and slightly increased performance of our learning algorithm. We thus plan to replace Table 1 in the final manuscript with these new results with $n=10$ independent runs each.

Using the open-loop steady state Jacobian as described removes the temporal dynamics of the feedback weights. However, the Jacobian is still input-dependent. Although our focus was not to solve the control problem, we acknowledge that related work used a similar feedback control mechanism (Meulemans 2021, 2022) and demonstrated learning with feedback weights that are not input-dependent and instead reflect the *average* transpose of the Jacobian across the dataset. To test whether our model would also support this static feedback weight regime, we performed the same simulations as in Table 1, but averaged the calculated feedback weights for each mini-batch. The results of these simulations are summarized in Table 2 of the attached PDF. Such feedback weights reflect the *average* Jacobian. They could be learned online from noise correlations, as shown by Meulemans et al. (2021).

Together these findings indicate that our model is robust to the details of the feedback weights. We plan to include these results in the final manuscript. Furthermore, we expect that existing online learning algorithms for the feedback weights could be adapted to our dis-inhibitory control model in the future and will discuss this possibility in the revised discussion section.

### 2. More comprehensive comparison of in-vitro slice experiments with previous models

Some reviewers were concerned that we did not clarify sufficiently how our experimental predictions differ from existing bio-plausible credit assignment models. We agree that we could have done a better job at describing the commonalities and differences.

To address this point, we repeated the same experiments for the microcircuits and associated learning rules presented in related papers on credit assignment. Specifically, we compare to dendritic errors rules, (PDF Fig. 1a–c; Gilra & Gerstner (2018), Urbanczik & Senn (2014), Sacramento et al. (2018)),deep feedback control (PDF Fig. 1d,e; Meulemans et al. (2021 & 2022)) and burst multiplexing (PDF Fig. 1f,g; Payeur et al. (2021), Greedy et al. (2022)).Most models cannot reproduce the experimentally observed BCM-like learning curve. While burst-multiplexing models could reproduce a learning rule with BCM-like moving threshold if output spikes are Poisson distributed, the proposed underlying mechanism requires a spatially well-separated apical dendrite, specific to cortical L5 neurons. It is, therefore, difficult to see how L2/3 cortical neurons or, for instance, neurons in the basolateral amygdala could use a similar mechanism. In both cases, however, prominent disinhibitory circuit motifs are known to exist in neurobiology. In reality it is entirely conceivable that both burst-multiplexing and inhibitory control work side-by-side. We will clarify these commonalities and differences by adding a discussion and similar comparison along with a summarizing figure in the supplementary material of our final manuscript.

### 3. Preliminary results on learning with more biologically plausible inhibitory connectivity

Some reviewers raised a concern about the plausibility of the one-to-one inhibitory connectivity. We agree with the reviewers that this connectivity is  biologically implausible. We made this simplifying model choice because it allowed us to directly relate the model to existing frameworks with neuron-specific error signals. Reducing the number of interneurons to reflect realistic ratios requires feedback to be low-dimensional which opens up new exciting research questions beyond the scope of the present article. We have started to empirically test the notion of learning with low-dimensional feedback and obtained promising preliminary results (see Figure 2 of the attached PDF). Specifically, we investigated training ANNs with low-rank feedback weights in a Direct Feedback Alignment setting  (Nøkland, 2016). While these results indicate that learning with low-dimensional feedback (and thus realistic number of interneurons) signals is possible, relating these findings to a recurrent architecture compromising excitatory and inhibitory neurons requires additional work that is beyond the scope of the current paper.

---

> ### Author Response · Authors · 2023-08-21
>
> We would like to thank the reviewers for their thorough review and constructive feedback. We appreciate the time dedicated to our paper and look forward to improving it based on the provided suggestions.

---

### Decision · Program_Chairs · 2023-09-21

**Decision:**

Accept (poster)

**Comment:**

This paper proposes a model of learning in cortical circuits that relies on Hebbian plasticity paired with disinhibition driven by error signals. The authors show that the model accounts for plasticity data both in the presence and absence of inhibition. They also show that the model compares comparably to backpropagation on some simple non-linear tasks. The model makes some specific predictions about inhibitory modulation of excitatory plasticity.

The reviewers felt that the paper was interesting, though they raised some concerns with respect to the novelty of the contributions, the robustness of the comparisons to experimental data, and some remaining issues of biological implausibiltiy. After rebuttals, the reviewers were largely satisfied, and all had scores above the acceptance threshold, so a decision of accept was reached.